# Selectivity matters: selective ROCK2 inhibitor ameliorates established liver fibrosis via targeting inflammation, fibrosis, and metabolism

Alexandra Zanin-Zhorov [1✉], Wei Chen[1,7], Julien Moretti [1,7], Melanie S. Nyuydzefe[1,7], Iris Zhorov[1], Rashmi Munshi[2], Malavika Ghosh[2], Cindy Serdjebi [3], Kelli MacDonald[4], Bruce R. Blazar[5], Melissa Palmer[6] & Samuel D. Waksal[1]

The pathogenesis of hepatic fibrosis is driven by dysregulated metabolism precipitated by chronic inflammation. Rho-associated coiled-coil-containing protein kinases (ROCKs) have been implicated in these processes, however the ability of selective ROCK2 inhibition to target simultaneously profibrotic, pro-inflammatory and metabolic pathways remains undocumented. Here we show that therapeutic administration of GV101, a selective ROCK2 inhibitor with more than 1000-fold selectivity over ROCK1, attenuates established liver fibrosis induced by thioacetamide (TAA) in combination with high-fat diet in mice. GV101 treatment significantly reduces collagen levels in liver, associated with downregulation of pCofilin, pSTAT3, pAkt, while pSTAT5 and pAMPK levels are increased in tissues of treated mice. In vitro, GV101 inhibits profibrogenic markers expression in fibroblasts, adipogenesis in primary adipocytes and TLR-induced cytokine secretion in innate immune cells via targeting of Akt-mTOR-S6K signaling axis, further uncovering the ROCK2-specific complex mechanism of action and therapeutic potential of highly selective ROCK2 inhibitors in liver fibrosis.

[1] Graviton Bioscience B.V, Amsterdam 1017 CG, Netherlands. [2] Aragen Bioscience Inc, Morgan Hill, CA 95037, USA. [3] Biocellvia, 13001 Marseille, France. [4] QIMR Berghofer Medical Research Institute, Brisbane 4006, Australia. [5] Division of Blood & Marrow Transplant & Cellular Therapies, University of MN, Masonic Cancer Center and Department of Pediatrics, Minneapolis, MN 55455, USA. [6] Liver Consulting LLC, Dix Hills, NY 11746, USA. [7] These authors contributed equally: Wei Chen, Julien Moretti, Melanie S. Nyuydzefe. ✉email: Alexandra.zanin-zhorov@gravitoncorp.com

Rho-associated coiled-coil-containing protein kinases (ROCKs) have been implicated in the pathology of a plethora of diseases characterized by chronic inflammation, excessive fibrosis, and metabolic dysregulation[1–4]. ROCK1 and ROCK2 isoforms share more than 90% homology within the kinase domain[5], however the intracellular function of these proteins is not redundant and vary between different cellular experimental systems[6]. While increased ROCK activity has been linked to inflammatory and autoimmune disorders through its involvement in regulation of cytoskeletal rearrangement[7,8], only the ROCK2 isoform has been shown to regulate secretion of pro-inflammatory cytokines including IL-17 and IL-21 via STAT3 and IRF4-dependent mechanisms in mice and humans[9–11]. Furthermore, pharmacologic ROCK2 targeting also upregulates STAT5 phosphorylation and regulatory T cell (Treg) suppressive function, suggesting that ROCK2 controls the balance between pro-inflammatory and immunosuppressive subsets of T cells both in vitro and in vivo[10,12–14]. Recently, ROCK2 has been implicated in the regulation of pro-inflammatory pathogenic macrophages differentiation via STAT3/cofilin signaling pathways in the context of liver fibrosis in mice[15].

While both ROCK isoforms are activated by small Rho GTPase in response to a variety of profibrotic signals[2], recent studies demonstrated isoform-specific functions in the regulation of cytoskeletal dynamics, activation of different downstream intracellular targets including myosin light chain, LIM kinase, cofilin and expression of key profibrotic genes, such as connective tissue growth factor (CTGF) and smooth muscle actin[16–18]. ROCK1 is critical for the formation of stress fibers, whereas ROCK2 mediates cortical contractility and phagocytosis of fibronectin in fibroblasts[19]. The opposite roles of ROCK1 (protective) and ROCK2 (pathological) were reported in cardiomyocytes during pressure-overload heart failure with postcapillary pulmonary hypertension in mice[20]. In diabetic animals ROCK2, but not ROCK1 contributes to regulation of TGF-β-induced profibrotic gene expression[16]. Importantly, Knipe et al. demonstrated that down-regulation of only one isoform, either ROCK1 or ROCK2, was sufficient to protect mice from bleomycin-induced pulmonary fibrosis in mice[21]. Thus, selectively targeting ROCK2 alone may have therapeutic advantage compared to inhibition of both isoforms in fibrotic diseases.

Nonalcoholic steatohepatitis (NASH), the severe form of nonalcoholic fatty liver disease (NAFLD), characterized by steatosis, hepatocyte injury, liver inflammation, and typically liver fibrosis, is an integral part of the metabolic syndrome. In fact, there is ongoing debate concerning changing the name of this disease to metabolic dysfunction-associated fatty liver disease (MAFLD)[22]. This underscores the importance of targeting both metabolic and fibrotic pathways as well as immune imbalances when developing new therapies for this medical condition with a high unmet clinical need[23,24]. Increased ROCK activity has been detected in tissues from obese, insulin-resistant animals as well as in leukocytes from patients with metabolic syndrome[25]. The ROCK-mediated regulation of insulin sensitivity appears to be isoform-dependent and tissue-specific: while ROCK1 is required for glucose transport in skeletal muscle[26] and negatively regulates insulin signaling in adipose tissue[27], ROCK2 promotes cardiac and adipose tissue insulin resistance in obese mice fed a high-fat diet (HFD)[28,29]. In addition, selective ROCK2, but not ROCK1/2 inhibition suppressed adipocyte differentiation in both murine 3T3-L1 cell and human adipose-derived stem cell (hADSC) cultures[30,31]. These data imply the importance of selective ROCK2 targeting in development of potent anti-metabolic agents.

The development of a single therapeutic agent with ability to concomitantly target metabolism, inflammation and hepatic fibrosis would be the "holy grail" approach to treat patients with NASH who have liver fibrosis. Here we characterize a highly ROCK2-specific inhibitor, GV101, which efficiently treats established liver fibrosis in mice. We further uncovered a previously unappreciated integral anti-inflammatory and anti-metabolic effect of ROCK2 inhibition in human monocytes and Kupffer cells, two types of innate immune cells that actively contribute to liver disease progression. Altogether, the present study underlines a complex mechanism of action of a novel and selective ROCK2 inhibitor that concurrently targets adaptive and innate immune cells as well as adipocytes and fibroblasts, achieving potent anti-inflammatory and anti-fibrotic effects, enabling efficient treatment of established liver fibrosis.

## Results

**Highly selective ROCK2 inhibitor GV101 alleviates established TAA-induced liver fibrosis.** Thioacetamide (TAA) is a commonly used toxic agent to induce and establish liver fibrosis in rodents. Previously, it was demonstrated that ROCK2 inhibition attenuates profibrogenic immune cell function to reverse TAA-induced liver fibrosis in mice. GV101 is orally available selective ROCK2 inhibitor, which is more than 1000-fold selective for the ROCK2 over ROCK1 isoform (Supplementary Fig. 1a). Previously published data was generated by using Belumosudil (formerly named – and hereafter referred to as KD025), a selective ROCK2 inhibitor which has about 100-fold selectivity for ROCK2 (Supplementary Fig. 1a). We compared the therapeutic potential of KD025 and GV101 to down-regulate established TAA-induced liver fibrosis in mice that were treated with TAA supplemented in drinking water for 9 weeks. The kinase inhibitors were administered by oral gavage for 3 weeks starting from week 7 to week 9 (Fig. 1a): GV101 was dosed at 30, 100, and 150 mg/kg/day and KD025 was dosed at 150 mg/kg/day. The quantification of Hydroxyproline (HYP), a non-proteinogenic amino acid synthesized by post-translational hydroxylation of proline during collagen biosynthesis, in the median lobe of the liver (Fig. 1b) and histological analysis using Sirius red staining (Fig. 1c and Supplementary Fig. 1b) showed that TAA administration for 6 weeks (TAA W6 group) induced advanced liver fibrosis characterized by a significant increase in collagen levels compared normal water drinking mice. There was a slight, but not significant elevation in HYP levels from 6 weeks to 9 weeks of TAA administration (TAA W6 and TAA W9 groups, respectively). Oral administration of GV101 beginning on week 6 resulted in dose-dependent reduction of liver fibrosis characterized by HYP assessment with statistical significance achieved at the highest dosing regimen (Fig. 1b). Notably, by normalizing the HYP data to week 6 (start of treatment) we found that the highest dose of GV101 (150 mg/kg) reduced collagen to levels observed in livers of normal water drinking mice suggesting that the highly selective ROCK2 inhibitor not only blocks the progression but reverses the established TAA-induced liver fibrosis in mice (Supplementary Fig. 1c). Even though the levels of KD025 in serum were more than 4-fold higher compared to GV101 (Supplementary Fig. 1d), the effect of KD025 treatment was less robust and did not reach the statistical significance (Fig. 1b), which can be attributed to a more limited potency of KD025 against ROCK2 (Supplementary Fig. 1a). In addition, GV101 treatment at 150 mg/kg significantly reduced alkaline phosphatase (ALP) levels and lowered the aspartate transaminase (AST)/alanine transaminase (ALT) ratio (Supplementary Fig. 1e), further suggesting improvement of liver function. Moreover, selective ROCK2 inhibition by both GV101 and KD025 markedly reduced the levels of phosphorylated (p)Cofilin (validated ROCK2 target involved in cytoskeletal remodeling during

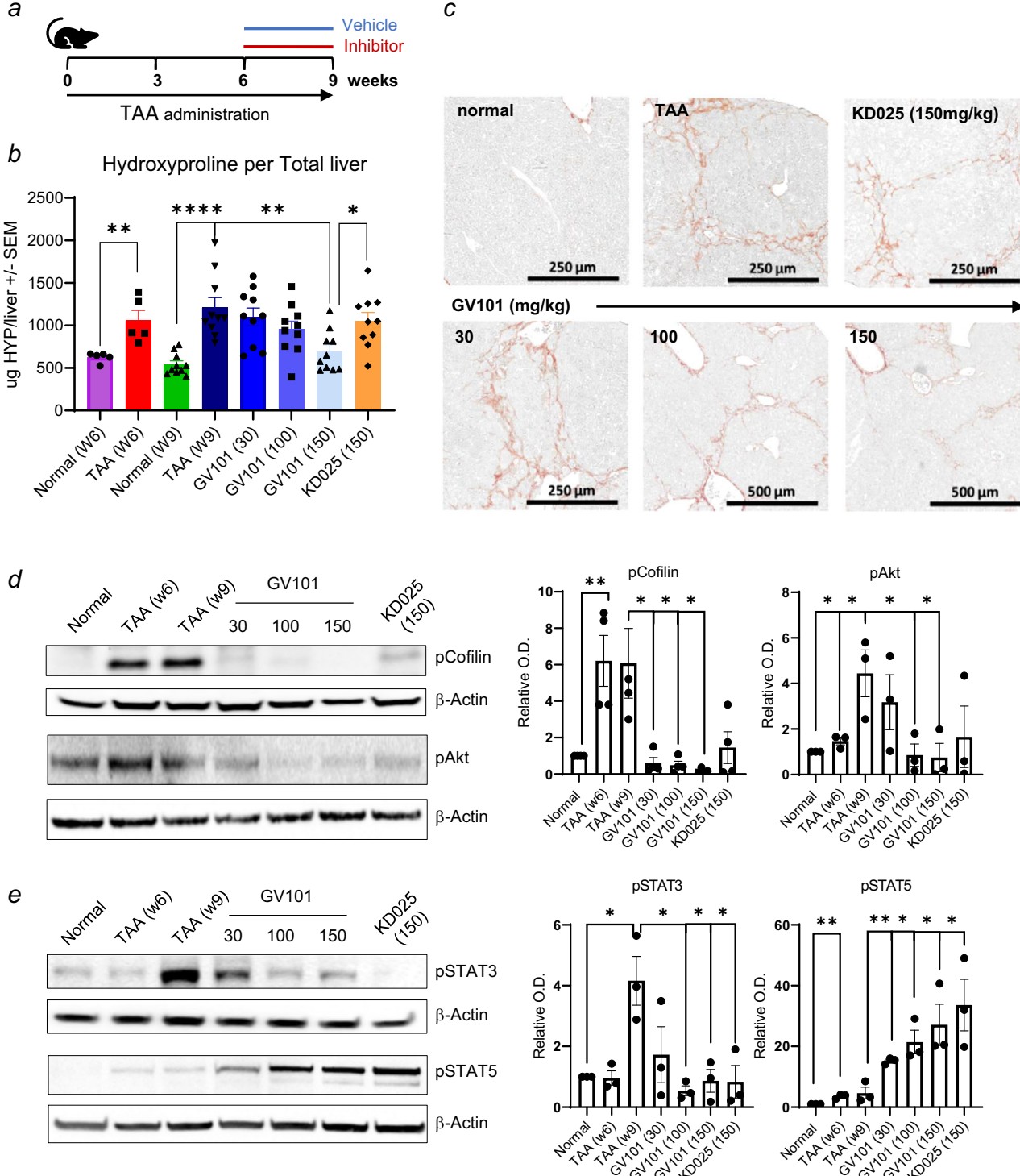

**Fig. 1 Selective ROCK2 inhibitor GV101 decreased established TAA-induced liver fibrosis. a** C57BL/6 female mice were treated with vehicle, GV101 (30, 100, and 150 mg/kg) or KD025 (150 mg/kg) by oral gavage on Week 6 after starting TAA in drinking water and continued for 3 weeks. **b** Hydroxyproline was assessed in the median lobe of liver collected on week 6 (normal W6 and TAA W6 with 5 mice/group) or harvest day on Week 9 (normal W9, TAA W9 and treatments with 10 mice/group). **c** Representative images of Picrosirius Red (PSR) staining are shown for each group. Liver (**d**) and spleen (**e**) tissue lysates were prepared in RIPA buffer and levels of pCofilin, pAkt, pSTAT3, and pSTAT5 were determined by Western Blot. Graphs represent mean +/− s.e.m. Unpaired t-test statistical analysis was performed: *$p \leq 0.05$; **$p \leq 0.01$; ***$p \leq 0.001$; ****$p \leq 0.0001$.

fibrosis), pAkt, pSTAT3 and concurrently up-regulated STAT5 phosphorylation in tissues including livers and spleens after 3 weeks of treatment compared to vehicle-treated animals (Fig. 1d, e) without changing the total levels of ROCK2 protein in the tissue (Supplementary Fig. 1f). These data confirm the target engagement of ROCK pathway for both inhibitors and further validate previously reported ROCK2-specific mechanism of action in vivo.

**GV101 concurrently targets fibrogenic, metabolic and inflammatory pathways in vivo**. Diet-induced obesity has been shown to play a role in development and progression of hepatic fibrosis by promoting lipogenic pathways and concomitant inflammation in liver[22]. Therefore, we combined TAA-supplemented drinking water with feeding mice a high fat, so called Western Diet (WD) for 12 weeks (Fig. 2a). To define the therapeutic potential, treatment with the selective ROCK2 inhibitors or vehicle started on week 6, when a significant increase in liver collagen levels was demonstrated by HYP assessment (Fig. 2b) and histological analysis (Fig. 2c and Supplementary Fig. 2a). Oral administration of GV101 for 6 weeks (week 6 to week 12) profoundly reduced collagen in the liver with significance achieved even at the lowest dose (30 mg/kg) of the inhibitor (Fig. 2b), which correlates with higher levels of GV101 in the serum compared to animals fed normal diet (Supplementary Fig. 2b). In contrast, 150 mg/kg dose of KD025 was not well tolerated leading to weight loss and death of two animals due to potential hepatotoxicities associated with the chemical structure of KD025. This led to the dose being reduced to 100 mg/kg from week 10 and continued for an additional two weeks until the end of the study without decreasing the liver collagen levels compared to vehicle treated animals (Fig. 2b). The body weight of animals treated with GV101 did not change significantly (Supplementary Fig. 2c), but serum sample analysis showed that reduction of collagen in the liver following GV101 administration was also associated with a robust decrease in levels of insulin and leptin, a key modulator of insulin secretion and sensitivity (Fig. 2d). There was also a mild, but significant reduction in total and LDL cholesterol levels compared to vehicle treated animals (Supplementary Fig. 2d). Moreover, selective ROCK2 inhibition significantly reduced blood levels of pro-inflammatory cytokines including IL-1β and IL-17 (Fig. 2d). KD025 administration resulted in mixed results (Fig. 2d) that can be attributed to toxicities observed in treated animals due to a very high level of the drug in the blood (Supplementary Fig. 2b). In line with the peripheral signs of improved metabolism and reduced inflammation, tissue analysis demonstrated that GV101 treatment significantly reduced levels of pCofilin, pSTAT3 and pAkt (inflammatory drivers), while the levels of pAMPK, a key regulator of glucose and fatty acid metabolism, were increased in both livers (Fig. 2e) and spleens (Supplementary Fig. 2e) of treated mice. The levels of ROCK2 expression in tissue did not change as a result of the treatment with inhibitors (Supplementary Fig. 2f). These data further demonstrate the potential of a highly selective ROCK2 inhibitor to reverse established liver fibrosis via combined targeting of fibrotic, inflammatory, and metabolic signaling pathways in vivo.

**GV101 inhibits adipogenesis in human subcutaneous pre-adipocytes and murine 3T3L1 cells**. To further elucidate the direct impact of selective ROCK2 inhibition on metabolism, fibrosis and inflammation, in vitro cell lines and primary cell cultures were employed and the effects of two selective ROCK2 inhibitors (GV101 and KD025) and pan-ROCK inhibitors, such as H-1152 and GSK429286 were compared (Supplementary Fig. 1a). Adipose tissue and its secreted adipokines are well known to induce the flux of lipids, create lipotoxic milieu and promote inflammation as well as fibrosis in the liver[32,33]. Human subcutaneous pre-adipocytes (Fig. 3a and Supplementary Fig. 3a) and murine 3T3L1 cells (Supplementary Fig. 3b) were induced to differentiate into adipocytes in the absence or presence of indicated ROCK inhibitors. Both ROCK2 selective inhibitors, GV101 and KD025, suppressed adipogenesis shown by Oil Red O staining (Fig. 3a and Supplementary Fig. 3a, b). Further quantification of lipid accumulation by measuring absorbance of

extracted Oil Red O from stained cells showed that GV101 and KD025 inhibited adipogenesis in a dose dependent manner in both human and murine cells (Fig. 3a and Supplementary Fig. 3c). GV101 showed a slightly less potent effect than KD025 but achieved comparable EC-50 in both human (4.5 μM for GV101 and 2.1 μM for KD025) and murine 3T3L1 cells (2.4 μM for GV101 and 1.1 μM for KD025). The potent anti-adipogenic effect of selective ROCK2 inhibition was associated with a dose dependent downregulation of Glut4 mRNA (Fig. 3b), which is a metabolic marker induced during adipogenesis[34]. In contrast, the pan-ROCK inhibitor H-1152 had no effect on adipogenesis in human cells (Fig. 3a), minimal effect on 3T3L1 cell cultures with an EC-50 of 28 μM (Supplementary Fig. 3c) and the mRNA level of Glut4 (Fig. 3b). These results are consistent with previous reports showing that KD025 inhibited adipogenesis in 3T3L1 cells[31] and human adipose-derived stem cells, while lipid accumulation was not reduced after exposure to several pan-ROCK inhibitors, which in contrast slightly enhanced adipogenesis in vitro[30]. Since the anti-adipogenic effect of KD025 was reported partially due to off-target inhibition of Casein Kinase 2α (CK2α)[35], we examined the capacity of GV101 to inhibit CK2α and found that 95% of CK2α activity remains intact in the presence of 10 μM of the inhibitor, with EC50 > 30 μM (Supplementary Fig. 3d). These data implicated that GV101-mediated down-regulation of adipogenesis is unlikely due to CK2α inhibition, whereas targeting of both CK2α and ROCK2 contributes to the anti-adipogenic effect of KD025.

AMPK is a key factor which senses nutritional and hormonal signals for adipogenesis process. The ROCK1/AMPK axis is involved in controlling metabolic functions in both liver and muscle[36,37], however whether ROCK/AMPK, especially ROCK2/AMPK pathway, contributes to regulating metabolic signaling pathways in adipose tissue is not known. By using ROCK1- and ROCK2-specific small interfering RNA (siRNA) that reduces expression of ROCK1 and ROCK2 in cells by 75% and 85%, respectively[10], we found that down-regulation of ROCK1 or ROCK2 increased AMPK phosphorylation (pAMPK) in human subcutaneous pre-adipocytes similar to treatment with 5-Aminoimidazole-4-carboxamide ribonucleotide (AICAR), a known stimulator of AMPK phosphorylation and activity (Supplementary Fig. 3e, f). Moreover, both selective ROCK2 inhibitors robustly up-regulated pAMPK, while the pan-ROCK inhibitor, H-1152, only had a minimal effect in human subcutaneous pre-adipocytes (Fig. 3c), consistent with its weak anti-adipogenic effect (Fig. 3a). These data show a previously unappreciated effect of ROCK2 targeting on regulation of adipogenesis associated with increased AMPK phosphorylation and further confirm the anti-metabolic potential of highly selective ROCK2 inhibitors.

**GV101 inhibits fibrogenic pathway in human fibroblasts**. A 30-fold increased potency of GV101 against ROCK2 compared to KD025 (Supplementary Fig. 1a) also resulted in profound direct anti-fibrotic effect as was demonstrated by using MRC-5 fibroblasts cells derived from human fetus lung and stimulated by TGF-β in vitro. First, we confirmed that TGF-β1 markedly induced phosphorylation of Cofilin (pCofilin), and expression of profibrotic markers such as α-SMA, CTGF, COL3A1 (Fig. 3d, e). Treatment with GV101 and KD025 down-regulated levels of pCofilin, α-SMA and Collagen 1 in a dose-dependent manner (Fig. 3d), which correlated with reduced mRNA levels of CTGF, COL3A1, α-SMA (Fig. 3e) under the same experimental conditions. It should be noted that GV101 is more potent inhibiting α-SMA mRNA and protein expression (Fig. 3d, e) as well as reducing secretion of COL1A1 detected by ELISA in TGF-β1-

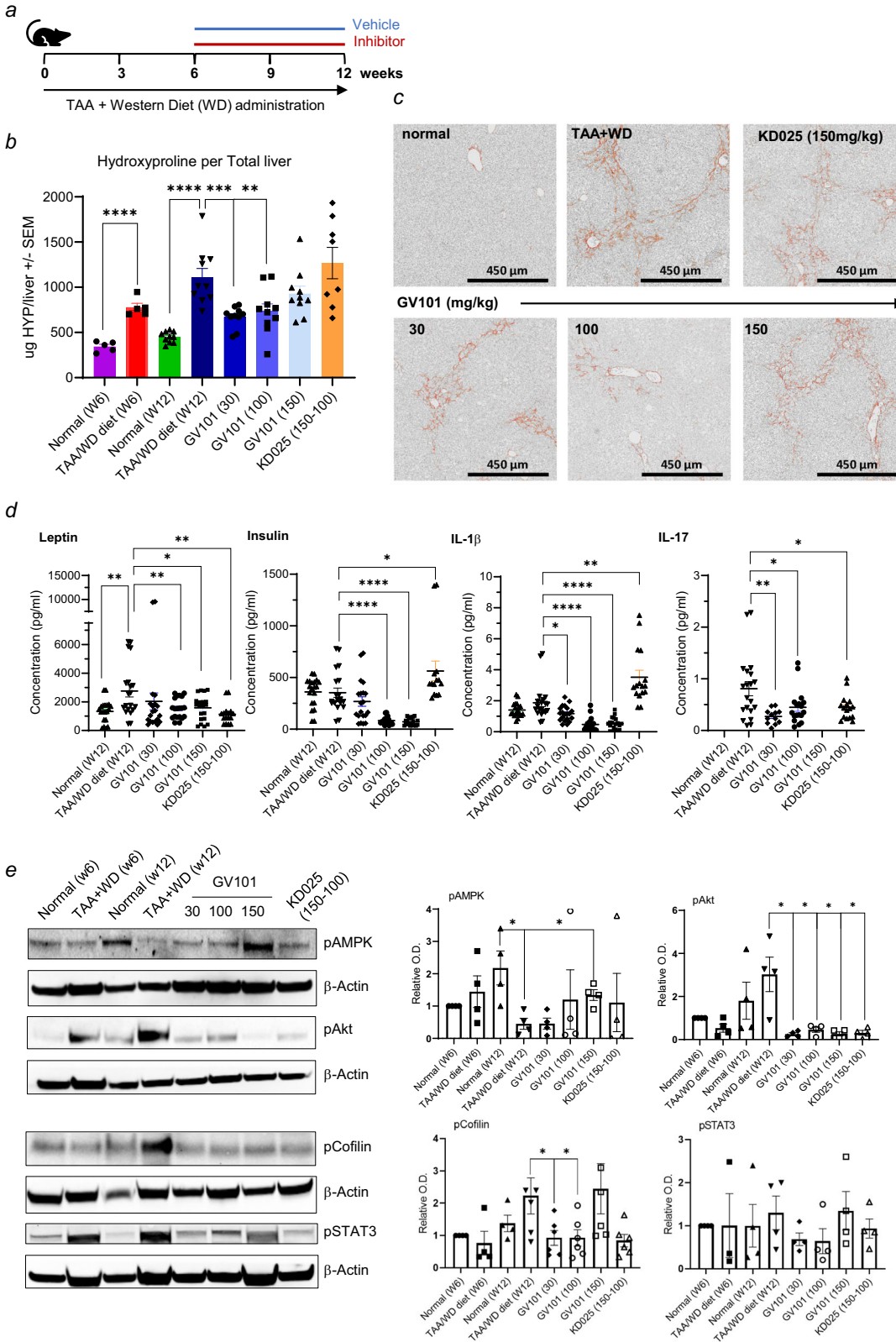

**Fig. 2 GV101 downregulates liver fibrosis induced by TAA in combination with Western Diet (WD) via targeting of inflammatory, fibrotic, and metabolic pathways. a** C57BL/6 female mice were treated with vehicle, GV101 (30, 100, and 150 mg/kg) or KD025 (150 mg/kg) by oral gavage on Week 6 after starting TAA in drinking water and continued for 6 weeks. **b** Hydroxyproline was assessed in the median lobe of liver collected on week 6 (normal W6 and TAA W6 with 5 mice/group) or harvest day on Week 12 (normal W9, TAA W9 and treatments with 10 mice/group). **c** KD025 dose was lowered to 100 mg/kg after 4 weeks of treatment due to toxicities. Representative images of Picrosirius Red (PSR) staining are shown for each group. **d** The levels of leptin, insulin, IL-1β and IL-17 in serum collected on Week 12 were measured by ELISA. **e** Liver tissue lysates were prepared in RIPA buffer and levels of pAMPK, pAkt, pCofilin, and pSTAT3 were determined by Western Blot. Graphs represent mean +/− s.e.m. Unpaired t-test statistical analysis was performed: *$p \leq 0.05$; **$p \leq 0.01$; ***$p \leq 0.001$; ****$p \leq 0.0001$.

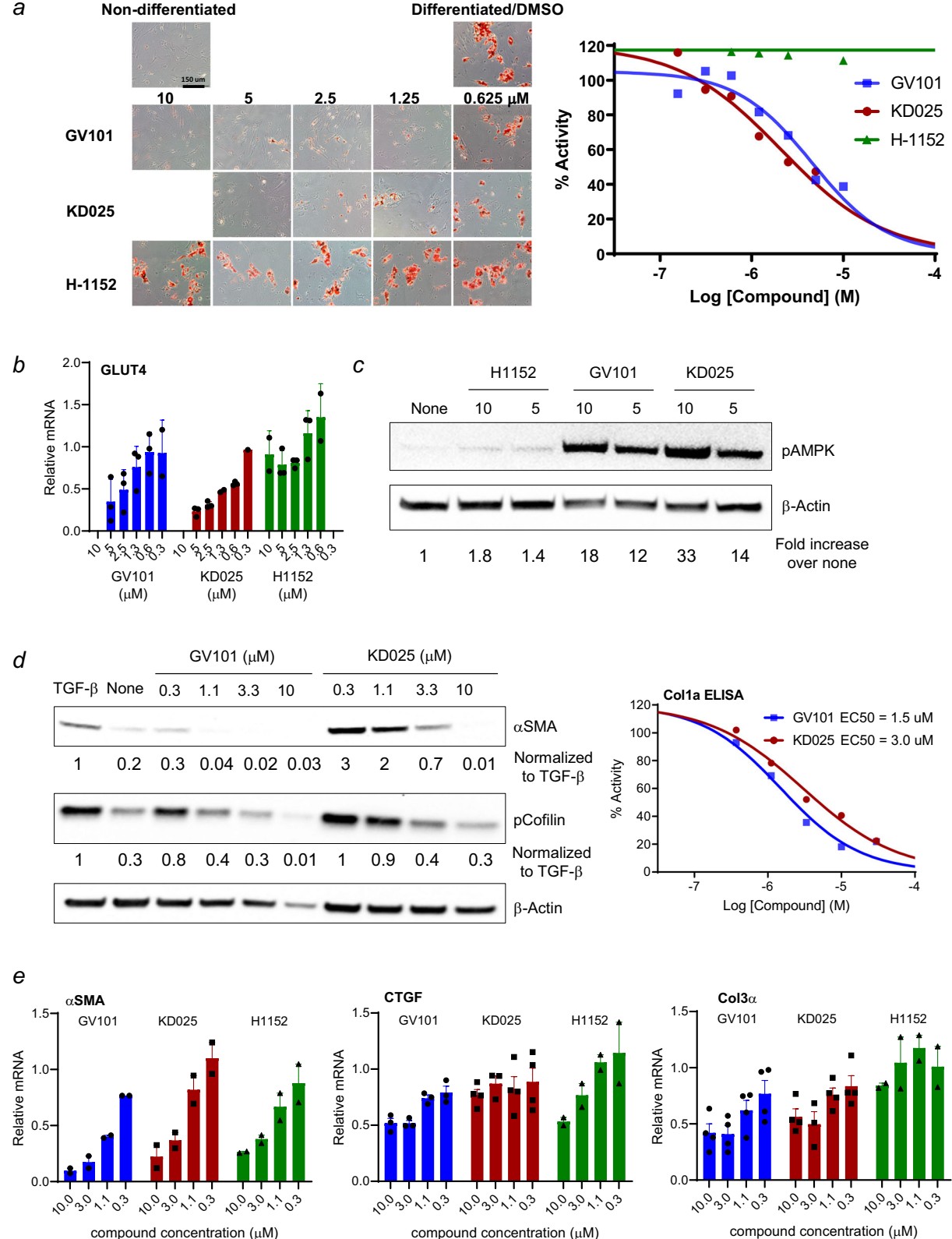

stimulated fibroblasts in comparison to KD025 (Fig. 3d). These in vitro findings correlate with the robust efficacy of GV101 to reverse established liver fibrosis in mice (Figs. 1, 2).

**Selective ROCK2, but not pan-ROCK inhibition down-regulates TH17 cytokine secretion.** Persistent inflammation is

considered to be the driving force for fibrogenesis in the liver via the interplay between different immune cell populations, including monocytes, Kupffer cells, natural killer (NK) cells and T cells[38]. Several studies have demonstrated the central role of IL-17-producing T cells (TH17 cells) in promoting hepatic inflammation and fibrogenesis through direct effects on Kupffer cells and hepatic stellate cells[39,40]. The novel highly selective ROCK2

**Fig. 3 GV101 and KD025 inhibit adipogenesis and fibrogenic pathways.** Human subcutaneous preadipocytes were induced adipogenesis with a medium containing differentiation cocktail (DM) for 7 days in the presence of indicated inhibitors. Differentiated cells were stained at day 8 by Oil Red O (**a**), quantification of lipid accumulation was measured by absorbance at 492 nM of extracted Oil Red O, all the absorbance was normalized to vehicle (DMSO/DM) treated differentiated cells (**a**), total RNA was extracted at day 8 and Glut4 gene expression was accessed by real-time RT-PCR (**b**). Human subcutaneous preadipocytes were treated with indicated inhibitors for 2 h before whole cell extracts were collected for Western blot analysis of phosphorylated AMPK and β-Actin (**c**). MRC-5 cells were pre-treated with various concentration of indicated inhibitors for 1 h before TGF-β1 stimulation for 48 h. Whole cell extracts were subjected to Western blot for the expression of α-SMA, pCofilin and β-actin (**d**). Western blots were quantified and normalized to the β-actin, and values are indicated under the corresponding immunoblots. Col1α levels in supernatants were measured by ELISA. All the Col1α levels were normalized with that of vehicle (DMSO/TGF-β1) treated cells (**d**), gene expression of α-SMA, CTGF, Col3α levels were measured by real-time RT-PCR. All the levels were first normalized with internal 18s control and then were normalized with the level of vehicle (DMSO/TGF-β1) treated cells (**e**). The data (**a**, **d**) are representative of three repeated experiments, (**b**, **e**) are average of three repeated experiments.

inhibitor GV101 profoundly decreased the secretion of IL-17, IL-21 and CXCL13 in human T cells stimulated by TH17-skewing conditions with average IC50s at 582 nM, 461 nM, and 830 nM, respectively (Supplementary Fig. 4a). The downregulation of cytokine secretion was accompanied by a dose-dependent downregulation of STAT3 phosphorylation (without reducing ROCK2 expression), reduction of the percentage of CXCR5[+]PD1[+] follicular T cells and concurrent upregulation of Foxp3[+] Tregs (Supplementary Fig. 4b–e, respectively) consistent with previously reported role of ROCK2 in regulation of TH17/Treg balance. The head-to-head comparison of GV101, KD025, and pan-ROCK inhibitors, H-1152 and GSK429286 showed that increased ROCK2 inhibition and selectivity over ROCK1 (Supplementary Fig. 1a) promoted robust downregulation of pro-inflammatory secretion in T cells as demonstrated by a shift in IC50s (Supplementary Fig. 4a).

**GV101 inhibits the pro-inflammatory response of human innate immune cells associated with liver homeostasis and fibrosis.** Recently, ROCK2-STAT3 pathway induced by IL-17 stimulation in murine bone marrow-derived macrophages (BMDM) was shown to promote a pro-inflammatory environment favorable to liver fibrosis in a TAA-induced liver fibrosis mouse model[15]. It is therefore of interest to better characterize the roles of ROCK2 pathway in innate immune cells and study its pharmacological targeting in the context of inflammatory and fibrotic diseases.

Several subsets of innate immune cells have been associated with liver homeostasis and disease. During disease, blood-circulating monocytes can infiltrate the inflamed liver, where they differentiate into monocyte-derived macrophages that complement and sometimes completely replace the initial pool of liver-resident macrophages: the Kupffer cells, a population of liver-specific tissue-resident macrophages (RTMs) from embryonic origin that represents the liver's first line of defense against infection or tissue damage[41,42]. Activation of monocytes or Kupffer cells triggers innate immune signaling pathways that result in the production of pro-inflammatory cytokines, inflammatory cell death, as well as recruitment and activation of specific adaptive immune cells. Therefore, any excessive and imbalanced pro-inflammatory response of liver-associated monocytes[41,43,44] or Kupffer cells[42,45] can lead to chronic liver inflammation and prolonged tissue damage that initiate a favorable basis for liver disease progression, notably during NASH-associated liver fibrosis[46,47].

We first assessed the effect of GV101 on the immune response of PBMCs after stimulation with LPS, ligand for the Toll-like receptor TLR4. Within the cell types that constitute PBMCs, TLR4 is highly expressed in myeloid cells – monocytes (10–30% of total PBMCs) – and to a much lower level in lymphoid cells – T cells (70–90%) and B cells (5–10%)[48,49]. PBMCs stimulated with LPS efficiently secreted the cytokines TNF, IL-23, IL-10, IL-1β and IL-6 (Fig. 4a and Supplementary Fig. 5a). We observed

that GV101 strongly inhibited in a dose-dependent manner the secretion of the pro-inflammatory cytokines TNF and IL-23, up to 40% and 70% reduction at 10 μM GV101, respectively (Fig. 4a). Secretion of pro-inflammatory IL-1β was modestly but significantly reduced upon GV101 treatment in a dose-dependent manner whereas reduction of IL-6 secretion only occurred at the highest (10 μM) concentration. (Supplementary Fig. 5a). In contrast, GV101 increased production of anti-inflammatory IL-10 in a dose-dependent manner, with a maximal increase of 50% at 10 μM (Fig. 4a). When comparing the effect of GV101 versus KD025 on ROCK2 targets in LPS-stimulated PBMCs, we observed that phosphorylation of Cofilin, pro-inflammatory STAT3 and anabolic Akt was reduced with GV101 treatment in a dose-dependent manner, and more efficiently than with KD025. Phosphorylation of catabolic AMPK, which was reduced upon LPS treatment, was increased by both GV101 and KD025, consistent with our observation in human subcutaneous pre-adipocytes (Fig. 4b and associated quantifications). As expected, inhibition of ROCK2 with GV101 or KD025 did not impact the total amount of ROCK2 protein in PBMCs (Supplementary Fig. 5b). Together, these results indicate that, upon LPS stimulation, selective inhibition of ROCK2 with GV101 efficiently impacts metabolic and pro-inflammatory targets and subsequently reduces the inflammatory response of PBMCs.

We then studied the effect of ROCK2 inhibition in myeloid innate immune cells. After purification from PBMCs, LPS-stimulated monocytes secreted various cytokines such as TNF, IL-23, IL-1β and IL-6 (Supplementary Fig. 5c). Consistent with our observations in PBMC, GV101 treatment significantly and dose-dependently reduced the production of pro-inflammatory cytokines TNF and IL-23 in monocytes (Supplementary Fig. 5c). We also observed a modest but significant reduction in highly pro-inflammatory inflammasome-dependent IL-1β (Supplementary Fig. 5c). GV101 inhibited both Cofilin and STAT3 phosphorylation in monocytes, as well as the Akt–mTOR–S6K anabolic pathway (Supplementary Fig. 5d) without affecting ROCK2 levels (Supplementary Fig. 5e). These results indicate that GV101-mediated selective inhibition of ROCK2 can inhibit both inflammatory and metabolic pathways specifically in monocytes, circulating innate immune cells, and reveal a novel mechanism by which GV101 can attenuate inflammation-related liver fibrosis.

We next assessed whether human primary liver-resident Kupffer cells – purified from patients – would respond to selective ROCK2 inhibition. TLR4 or TLR2/TLR6 stimulation, respectively with LPS or Pam2CSK4, induced notable secretion of IL-6 and TNF by Kupffer cells (Fig. 4c and Supplementary Fig. 5f). Selective ROCK2 inhibition with GV101 led to a robust and dose-dependent reduction of both cytokines' levels. Cytokine secretion was even impaired with 5 μM GV101, with 94% reduction for IL-6 and 83% reduction for TNF in response to LPS, and 75% reduction for both IL-6 and TNF in response to Pam2CSK4 (Fig. 4c and Supplementary Fig. 5f), indicating that

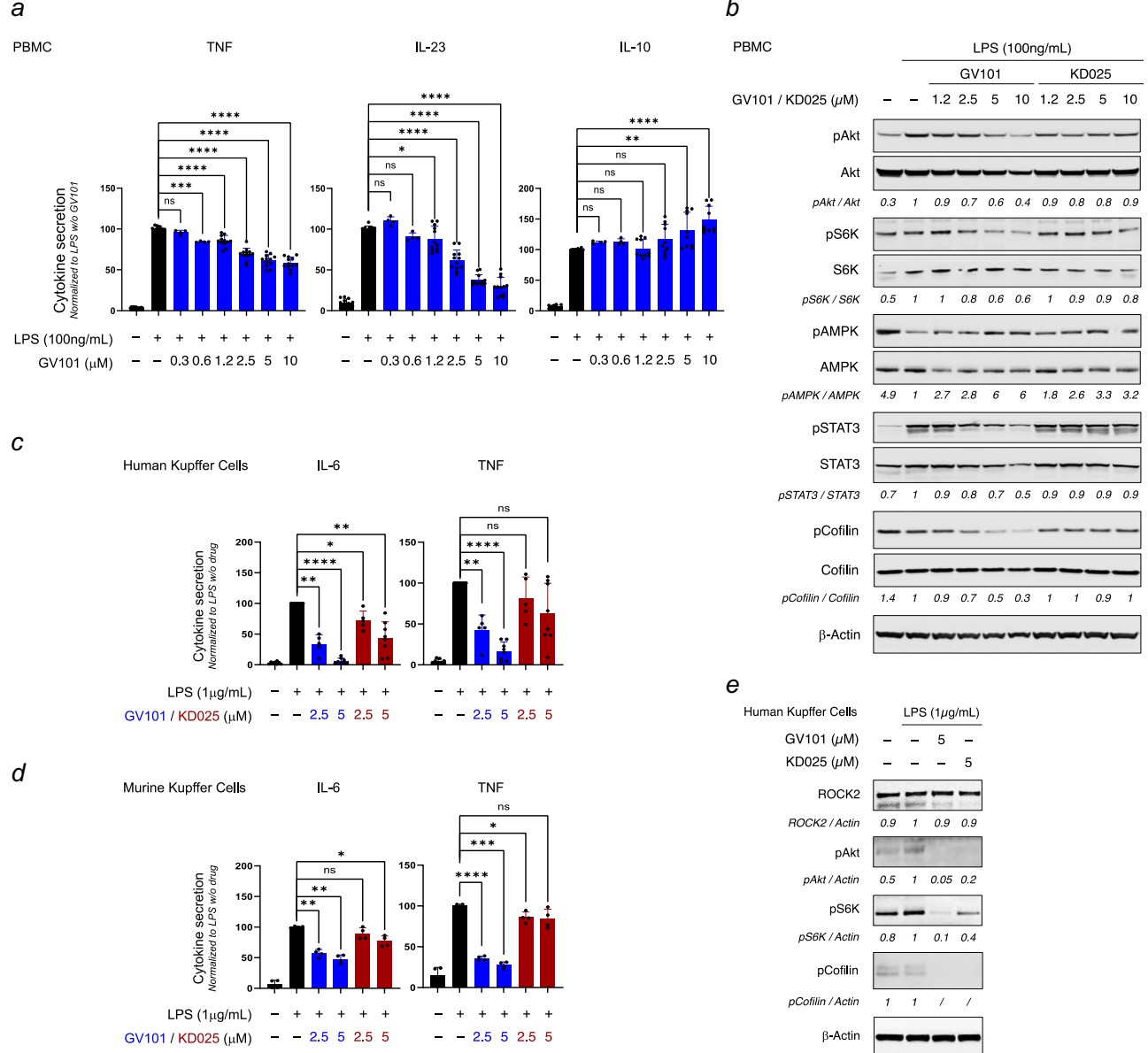

**Fig. 4 GV101 inhibits the pro-inflammatory response of primary PBMCs and Kupffer Cells in mouse and human.** PBMCs were treated with indicated doses of GV101 or KD025 1.5 h before stimulation with LPS. After 24 h, TNF, IL-23, and IL-10 secretion in supernatants was analyzed by ELISA and normalized ($n = 12$ except GV101 0.3 and 0.6 µM: $n = 4$) (**a**) and Western blots for the indicated proteins were performed on whole cell extracts (**b**). Human (**c, e**) or murine primary Kupffer cells (**d**) were treated with indicated doses of GV101 or KD025 1.5 h before stimulation with LPS and 24 h before collection of Kupffer cells supernatants for analysis of IL-6 and TNF secretion by ELISA followed by normalization (**c**: $n = 8$ except GV101/KD025 2.5 µM, $n = 5$; **d**: $n = 4$), and preparation of human Kupffer cell extracts for Western blots analysis of the indicated proteins (**e**). All experiments represent at least 4 independent repeats. Graphs represent mean $+/-$ s.e.m. In (**a, c, d**), one way ANOVA was performed followed by multiple comparisons Sidak tests to allow two-by-two comparisons. ns: not significant, $*p \leq 0.05$; $**p \leq 0.01$; $***p \leq 0.001$; $****p \leq 0.0001$. In (**b, e**), Western blots were quantified and normalized to the LPS-stimulated conditions, and values are indicated under the corresponding immunoblots.

human Kupffer cells are very responsive to ROCK2 selective inhibition. We also noted that similar doses of KD025 led to a milder inhibition of cytokine secretion compared to GV101, reaching significance only for IL-6 production in response to LPS with a 55% reduction after 5 µM KD025 treatment (Fig. 4c and Supplementary Fig. 5f). Similar observations were made in murine primary Kupffer cells: 5 µM GV101 reduced LPS- and Pam2CSK4-induced IL-6 and TNF secretion by at least 52% and 62% respectively, while 5 µM KD025 reduced IL-6 and TNF secretion by at best 37% and 25% respectively (Fig. 4d and Supplementary Fig. 5g). When assessing signaling pathways downstream of LPS in human Kupffer cells, we noted that

GV101 strikingly inhibited phosphorylation of anabolic Akt and S6K, as well as ROCK2 target Cofilin, without affecting ROCK2 levels (Fig. 4e). KD025 also strongly impacted Cofilin phosphorylation but had a lower effect than GV101 on Akt and S6K phosphorylation. All together, these observations demonstrate that selective ROCK2 inhibition results in a strong, pan-specie downregulation of Kupffer cells pro-inflammatory response, newly characterizing liver-resident macrophages as a target cell type for ROCK2 inhibition. Of note, when we assessed monocyte-derived macrophages – widely used as a model for different macrophages subsets including liver-resident macrophages – we observed once again that GV101 can impact ROCK2 targets

Cofilin, anabolic Akt-mTOR and catabolic AMPK pathways as it did in PBMCs, monocytes or Kupffer cells but to a lower extent (Supplementary Figs. 5h, i), suggesting that the effect of ROCK2 inhibition in innate immune cells is cell type-dependent and may impact more strongly specific tissue-resident macrophages subsets, such as Kupffer cells, which ontogenically and physiologically differ from monocyte-derived macrophages[44].

To the best of our knowledge, these results demonstrate for the first time that, along with the efficacy on adaptive immune cells and adipocytes, selective ROCK2 inhibition exerts an anti-inflammatory effect on several innate immune cells that have a predominant role in liver disease and fibrosis: liver resident Kupffer cells and monocytes that infiltrate the liver. Our findings uncover a complex mechanism of action of selective ROCK2 inhibition, which concomitantly targets metabolic, fibrotic, and inflammatory pathways in various cell types, including adaptive and innate immune cells, fibroblasts and adipocytes. They emphasize a high therapeutic potential for selective ROCK2 inhibitors in diseases that involve inflammation and fibrosis not only in the liver, but also in organs such as kidneys or lungs.

## Discussion

The development of a single therapeutic agent with ability to concomitantly target metabolism as well as hepatic inflammation and fibrosis would be the "holy grail" approach for effective treatment of patients with NASH who have liver fibrosis, since all three processes actively contribute to disease progression[50,51]. Dysregulated metabolism of cholesterol, sugars, and insulin resistance promote liver injury and fibrosis by increasing hepatic lipogenesis and triggering inflammatory pathways[52]. The uncontrolled activation of immune cells such as TH17 cells and macrophages induces the secretion of pro-inflammatory cytokines that drive the differentiation of hepatic stellate cells into activated fibroblasts leading to excessive accumulation of collagen, development of scar tissue and formation of fibrosis in the liver[22]. In this study, we report that a highly selective ROCK2 inhibitor, GV101, attenuates established liver fibrosis in mice induced by chemical injury alone (Fig. 1b) or in combination with a high fat, so called Western diet (Fig. 2b). In both models, collagen levels were reduced significantly compared to vehicle treated animals and the collagen amount observed in livers of GV101-treated mice were comparable to those detected in normal water drinking group. This suggests that the highly selective ROCK2 inhibitor not only arrests the progression but promotes the reversal of existing liver fibrosis in vivo. The efficacy of GV101 in both models correlates with its peripheral exposure: the concentrations of the inhibitor detected in serum of high fat diet fed mice and treated with 30 mg/kg of GV101 were comparable to levels detected in the 150 mg/kg group receiving normal diet (Supplementary Fig. 2b) and was associated with a similar degree of collagen reduction observed in the livers of treated mice in both groups (Figs. 1b, 2b). These data suggest a potential food effect on peripheral exposure of GV101 that was previously reported for Belumosudil (generic name of KD025)[53]. The serum levels of KD025 in high fat diet fed mice were 5X of levels achieved in patients (Belumosudil label at fda.gov) potentially causing toxicities observed in this group characterized by weight loss, death of treated animals and demonstrated mixed results with regards to pro-inflammatory cytokine assessment. KD025 treatment only mildly reduced levels of TAA-induced collagen levels in liver, which may be explained by limited potency of KD025 against ROCK2 (Supplementary Fig. 1a). Thus, potent ROCK2 inhibition is required to ameliorate established liver fibrosis induced by chemical injury in the settings of metabolic dysregulation, which is reminiscent of patients with NASH.

Tissue and serum analysis revealed that treatment with GV101 down-regulated pro-inflammatory signaling pathways such as STAT3 and Akt in the liver (Fig. 2e) associated with reduced levels of IL-17 and IL-1β in the serum (Fig. 2d). As expected, selective ROCK2 inhibition significantly reduced cofilin phosphorylation that is markedly increased in fibrotic livers (Figs. 1d, 2e). This further validates the role of the ROCK2 pathway in perpetuating fibrogenesis in vivo. The beneficial effects of ROCK2 targeting on metabolism in vivo were demonstrated by diminished levels of serum leptin, insulin and cholesterol associated with a robust increase in levels of tissue pAMPK in mice treated with a highly selective ROCK2 inhibitor (Fig. 2d, e; Supplementary Fig. 2d, e). Interestingly, the efficacy of GV101 treatment in TAA+Western Diet achieved plateau at 30 mg/kg dose in some of the measured readouts including Hydroxyproline, pCofilin, pSTAT3 and IL-17, whereas other parameters were modulated in a dose-dependent manner, such as Leptin, Insulin, IL-1β, pAMPK and pAKT. This might reflect the difference in dynamics, nature and localization of the measured readouts. Leptin can suppress both hepatic glucose production and lipogenesis at earlier stages of hepatic steatosis, however prolonged hyperleptinemia promotes pro-inflammatory and profibrogenic cascades leading to development of advanced liver fibrosis[54]. Additionally, peripheral insulin resistance and subsequent hyperinsulinemia have been shown to be factors that worsen liver fibrosis in preclinical studies across multiple models[32]. Therefore, the newly discovered triple effect of selective ROCK2 inhibition to target convergently pro-inflammatory, fibrotic, and metabolic pathways in vivo underscores the integral role of ROCK2 signaling in regulation of liver fibrosis.

AMPK plays a key role as an intra-cellular energy sensor improving insulin sensitivity in insulin-sensitive tissues like liver, skeletal muscle, and adipose tissue. When activated, AMPK leads to ATP production by shutting down anabolic pathway such as lipid and cholesterol synthesis, while enhancing catabolic pathways such as fatty acid oxidation[55]. Previous studies have indicated that ROCK1 negatively regulates AMPK phosphorylation and activity in the liver, skeletal muscle, and pancreatic β-cells via an unknown mechanism[25,36,37]. However, the role of ROCK2 in regulation of the AMPK pathway, especially in metabolic tissues, is less characterized. We found that pharmacological targeting with selective ROCK2 inhibitors or siRNA-mediated inhibition of ROCK2 resulted in a robust increase of AMPK phosphorylation (pAMPK) in human subcutaneous pre-adipocytes (Fig. 3c and Supplementary Fig. 3e), which mirrors the induction of pAMPK detected in both liver and spleen tissues of mice treated with GV101 or KD025 (Fig. 2e and Supplementary Fig. 2e). These data provide previously unknown evidence and establish a negative role of ROCK2 in regulating AMPK signaling. It is well documented that activated AMPK downregulates white adipogenesis through concomitantly suppressing clonal expansion of pre-adipocytes and inhibiting PPARγ and C/EBPα at early phase of adipogenesis[56]. Our results showed that two selective ROCK2 inhibitors decreased adipogenesis in both human and murine pre-adipocytes cultures defined by Oil Red O staining and Glut4 expression (Fig. 3a, b; Supplementary Fig. 3a–c). Consistent with previously published data[30,31], we found that pan-ROCK inhibition had minimal effect on adipogenesis (Fig. 3a; Supplementary Fig. 3a–c) as well as AMPK phosphorylation in vitro (Fig. 3c), further demonstrating different consequences of selective vs pan-ROCK inhibition in cells. While others proposed that the anti-adipogenic effects of KD025 are solely due to off-target inhibition of CK2, we ruled out this possibility by demonstrating that selective and highly potent ROCK2 inhibitor GV101 does not inhibit CK2 (Supplementary Fig. 3d), but markedly suppresses adipogenesis. Besides acting on AMPK pathway, we also observed

that selective-ROCK2 inhibitors robustly down-regulated Akt phosphorylation in both in vivo and in vitro settings (Figs. 1d, 2e, 4b, d), which is in line with the previously published data generated by using KD025 in 3T3L1 cells[31]. On the contrary, pan-ROCK inhibitors or genetic ablation of ROCK1 were reported to up-regulate Akt phosphorylation via PTEN-dependent mechanism[57,58]. The opposing effects on Akt signaling might explain the lack of effect of pan-ROCK inhibition on adipogenesis as well as pro-inflammatory cytokine secretion in T cells (Supplementary Fig. 4a).

ROCK2 signaling controls the balance between pro-inflammatory and immunosuppressive subsets of T cells, however the role of ROCK kinases in innate immune cells is less characterized. Both isoforms play a role in the polarization of macrophages M1 and M2 subsets, where ROCK2 induces fibrogenic M2-like macrophages associated with the development of age-related macular degeneration[59]. Importantly, recent study indicates that ROCK2 promotes liver fibrosis through its effects on profibrogenic macrophages[15]. We discover here that ROCK2 targeting impacts the innate immune response via integral effects on signaling associated with fibrogenic, inflammatory and metabolic pathways. The treatment of PBMCs with selective ROCK2 inhibitors reduced the levels of pro-inflammatory cytokines IL-6, TNF, IL-23 while increasing the levels of anti-inflammatory IL-10 upon LPS stimulation. This was associated with inhibition of inflammatory STAT3, and switch from anabolic (Akt-mTOR-S6K) to catabolic (AMPK) pathway (Fig. 4b). The effect of GV101 on PBMCs signaling pathways was more robust compared to KD025 treatment and correlated with more potent inhibition of ROCK2 (Supplementary Fig. 1a). Similar to PBMCs (Fig. 4a and Supplementary Fig. 5a), purified primary human monocytes (Supplementary Fig. 5b) and primary human and murine liver-resident Kupffer cells (Fig. 4c and Supplementary Fig. 5d) exhibited reduced pro-inflammatory cytokines secretion and metabolic switch with GV101 treatment upon TLR stimulation. These data underscore a potent anti-inflammatory effect of selective ROCK2 inhibition on different subsets of innate immune cells in vitro that might play a role in progression of fibrotic liver disease in vivo. As innate immune cells, monocytes, Kupffer cells and monocyte-derived macrophages can detect both foreign molecules associate with microbial infection (Pathogen-Associated Molecular Patterns (PAMPs)) and a category of self-molecules released upon cellular damage or tissue injury (Damage-Associated Molecular Patterns (DAMPs)). Detection is mediated by a variety of Pattern Recognition Receptors (PRRs) that notably include surface receptors such as TLRs or cytoplasmic receptors like NOD-like receptors (NLRs), which initiate immune signaling cascades[60–63]. ROCK kinases are not directly involved downstream of the TLR-dependent MyD88- or TRIF-mediated signaling cascades engaged with LPS or Pam2CSK4 stimulation[64], however they may regulate cytokine response via at least two possible mechanisms. First, mTOR pathway has been implicated in the regulation of TLRs signaling[65,66], therefore the GV101-dependent inhibition of Akt-mTOR-S6K axis we observe may contribute to inhibition of TLR-induced cytokine secretion in myeloid cells. Alternatively, the GV101-dependent inhibition of STAT3 we observed with GV101 may also reduce cytokine secretion by inhibiting JAK/STAT autocrine/paracrine inflammatory pathways downstream of cytokine receptors that amplify the cytokines secretion[67,68].

In conclusion, our data demonstrate the complex mechanism underlying the reversal of established liver fibrosis by selective inhibition of ROCK2 in vivo. By acting on immune, fibrotic, and metabolic pathways, in addition to targeting distinct cell types, the novel and highly selective ROCK2 inhibitor, GV101, tackles the disease on numerous different angles and turns out to be a very promising therapeutic approach for patients with liver fibrosis due to NASH, as well as for other diseases that progress on inflammatory, metabolic, and fibrotic grounds.

## Methods

**Inhibitors**. Selective ROCK2 inhibitors: KD025 (Selleck Chemicals LLC, #S7936) and GV101 (Beijing Tide Pharmaceuticals) and pan-ROCK inhibitors: H-1152 (Tocris Bioscience) and GSK429286 (MedChemExpress) were dissolved in DMSO and stored at −20 °C until used in cell-based assays. In addition to ROCK1, GV101 at 10 μM concentration was found to have no more than 50% inhibition of 50 intracellular kinases: MuSK(h), DRAK1(h), ALK2(h), LKB1(h), PDK1(h), TrkA(h), Fyn(h), CLK1(h), MSK1(h),PASK(h), Aurora-A(h), Flt3(h), p70S6K(h), TYK2(h), LIMK1(h), SAPK2a(h), activated IGF-1R(h), MLCK(h), Rsk1(h), AMPKα2(h), DYRK1B(h), GSK3α(h), JAK1(h), MRCKα(h), KDR(h), PKBα(h), Haspin(h), CK2α1(h), PDGFRα(h), DDR1(h), mTOR/FKBP12(h), PI3 Kinase (p110a/p85a)(h), CK2α2(h), LOK(h), MSSK1(h), IR(h), AMPKα1(h), GRK2(h), PKCα(h), JAK3(h), MRCKβ(h), PKCμ(h), Blk(h), PKG1β(h), Ron(h), Itk(h), PKA(h), ZAP-70(h), PKBγ(h), PKCθ(h), DMPK(h), Rsk2(h), Syk(h), PRK2(h), Fms(h), PKAcβ(h), Lck(h).

**Mice**. Female C57BL/6 mice six to seven weeks of age were obtained from Charles River. Animals were housed in a temperature-controlled room with a light/dark cycle of 12 h and with *ad libitum* access to food and water throughout the study. On the day of the study start, the animals were divided into two main groups, with one group ($n = 15$) receiving a normal drinking water and normal chow and the other group ($n = 75$) receiving TAA (Sigma Cat# 163678) supplemented water alone for 9 weeks or in combination with Western Diet (WD) chow for 12 weeks (21.1% fat, 41% Sucrose, and 1.25% Cholesterol by weight) purchased from Teklad diets (TD; #12052). At the beginning of week 7, 55 animals provided with TAA- supplemented water alone or in combination with WD were distributed into 5 treatment groups of ten animals each. Five animals from normal drinking water/chow group and TAA-water/TAA with WD group were harvested on week 6 before treatment initiation. On week 7 post randomization, animals were administered with indicated doses of GV101, KD025 or vehicle (0.5%CMC-Na, 800–1200) once daily via oral route for three weeks (week 7–week 9) or six weeks (week 7–week 12). Animals were harvested on week 9 after provided TAA supplemented water or week 12 after provided TAA supplemented water and WD. At end point, serum and tissues were collected, snap frozen or fixed, and fibrosis-related symptoms were analyzed. Individual animals were monitored daily for clinical observations, including general activity levels and morbidity. Any signs of discomfort were documented. Body weight was recorded twice weekly throughout the study period. This study was performed at Aragen Bioscience under the Animal Use Protocol: AUP#: 18-0803-MR. We have complied with all relevant ethical regulations for animal testing.

**Tissue and serum analysis**. Hydroxyproline analysis was performed by using median liver lobes snap frozen at harvest. The Western blot assessment was performed by using total spleen and caudate liver lobe tissues. The left liver lobes were fixed in 10% neutral buffered formalin and paraffin-embedded blocks were processed for histological analysis. For each tissue, 2 series of 2 sections (5 μm thick) were cut longitudinally and spaced by 50 μm. The two sections were laid on a same slide, stained with picrosirius red (PSR). Slides were scanned using the NanoZoomer-SQ scanner (Hamamatsu), at the magnification of

X20 (0.452 µm/pixel) and digital slides of whole sections captured using NDP.view 2 Hamamatsu software. The drug levels were measured in the right liver lobes and serum. Cytokine and liver enzyme analysis was performed by using serum samples.

**Isolation and stimulation of human primary immune cells: PBMCs, T cells, monocytes and monocytes-derived macrophages**. All human primary immune cells were purified from the peripheral blood (Leukopak) of healthy human donors between ages of 16 and 75 years (New York Blood Center, NY) by centrifugation at 1200 g for 20 min without brake over a Ficoll cushion. Human T cells were purified using RosetteSep™ Human CD4+ T cell Enrichment Cocktail (StemCell Technologies #15062), Vancouver, BC, Canada) before centrifugation, as previously described[10]. Human primary monocytes were isolated using RosetteSep Human Monocytes Enrichment Cocktail (StemCell Technologies #15068) before centrifugation. PBMCs were isolated directly by similar centrifugation. Purified PBMCs, T cells or monocytes were washed and centrifuged 2 times at 300 g and one last time at 100 g for 10 min without brake to help remove platelets. Immune cells were finally resuspended in RPMI containing 10% FBS and 1% Penicillin/Streptomycin (the immune cells medium) and counted.

For Th17-skewing activation, freshly purified CD4+ T cells were cultured in immune cells medium at a final concentration of $2 \times 10^6$/ml and stimulated with anti-CD3 mAb (5 µg/ml; eBioscience, San Diego, CA) and anti-CD28 mAb (5 µg/ml; eBioscience, San Diego, CA) in combination with IL-1β (50 ng/ml; R&D Systems Inc, Minneapolis, MN) and TGF-β (5 ng/ml; R&D Systems Inc, Minneapolis, MN) with or without indicated doses of ROCK inhibitors. Cytokine secretion was determined by ELISA after 48 h by using IL-17, IL-21 and CXCL13 kits. The percentage of Foxp3+ and CXCR5+PD1+ T cells was defined by Flow cytometry as previously described[10].

For experiments in PBMCs or monocytes, cells were kept resting for 12–16 h at 4 °C and then replated in appropriate non-tissue culture treated 24 W or 6 W plates at a concentration of 250,000 cells/cm$^2$/250 µL in immune cells medium before treatment and stimulation.

For differentiation of monocytes-derived macrophages (MDM), monocytes were directly plated in tissue culture treated 6 W plates at the same concentration in immune cells medium supplemented with 1% sodium pyruvate and 50 ng/mL human M-CSF (macrophage colony-stimulating factor, Peprotech #300-25) then differentiated for 7 days with medium replenishment every 2 days.

For TLR stimulation of PBMCs, monocytes or differentiated MDM, cells were pre-treated with indicated doses of GV101 or KD025 for 1.5 h in fresh medium. Cells were then stimulated with the TLR4 agonist LPS at 100 ng/mL (Sigma Aldrich #L6011). 24 h after stimulation, cells supernatants were collected, secreted cytokines were measured by ELISA and cells were lysed in Tris 50 mM pH 7.9, 300 mM NaCl and 1% Triton X-100 for preparation of whole cell extracts and western blot analysis.

**Culture and stimulation of human and murine primary Kupffer cells**. Frozen and previously characterized human primary Kupffer cells were obtained from Sekisui/Xenotech (#HK1000.H15). At the time of the study, human Kupffer cells were available from 5 donors: #H1292 (male donor, used for 2 independent repeats), #H1294 (male donor, used for 3 independent repeats), #H1467 (female donor, used for 1 independent repeat), #H1468 and #H1469 (male donors, each used for 1 independent repeat). Cells were thawed, plated in tissue-culture treated 48 W plates at a concentration of 125,000 cells/cm2/

250 uL in OptiThaw Kupffer Cell medium (Sekisui/Xenotech #K8700), and cultured for 10 days with medium replenishment every 3 days. Cells were pre-treated with indicated doses of GV101 or KD025 for 1.5 h in fresh medium. Cells were then stimulated with the TLR4 agonist LPS (Sigma Aldrich #L6011) or the TLR2/TLR6 heterodimer agonist Pam2CSK4 (Invivogen # tlrl-pm2s-1), both at 1 ug/mL, for 24 h before collection of supernatants, analysis of cytokines secretion by ELISA, and preparation of whole cell extracts. ELISA data represent 5-8 independent repeats from the 5 available Kupffer cells donors. For Western Blot analysis, and due to limited amount of cells for each sample, cells from donors #H1467, #H1468 and #H1469 were pooled to reach a sufficient amount of proteins in the whole cell extracts.

Murine primary Kupffer cells were obtained from Glow Biologics (#GBP-1758KC). At the time of the study, cells were available from 2 mice/lots: #14162122 and #17392123. Cells were plated and stimulated as detailed for human Kupffer cells, using immune cells medium.

**Pre-Adipocytes culture and adipogenesis assays**. Human subcutaneous pre-adipocytes (SP-F-1) and murine 3T3L1 (SP-L1-F) cells were purchased from ZenBio Inc. Cells were cultured in DMEM containing 10% fetal bovine serum, 100 U/mL of penicillin, 100 µg/mL of streptomycin and 1xAmphotericin B according to manufacturer's instruction. For human cell adipogenesis induction, cells were plated at confluence for overnight, the following day, cells were fed with adipocyte differentiation medium (DM-2, ZenBio) with or without indicated concentrations of ROCK inhibitors, 7 days after induction, cells were subjected to Oil Red O staining, or collected for RNA extraction, or collected for western analysis. For murine 3T3L1 cell adipogenesis, cells were plated at confluence, after two days, cells were fed with differentiation medium (DM-2, ZenBio) with or without indicated concentrations of ROCK inhibitors, incubated for 3 days, then cells were switched to adipocyte maintenance medium (AM-1, ZenBio) with or without indicated concentrations of ROCK inhibitors for 3 more days. 7 days after induction, cells were subjected to Oil Red O staining, or collected for RNA extraction, or collected for western analysis.

**Oil Red O staining**. Cells were washed with 2xPBS, fixed with 10% formalin for 1 h, then washed with $2 \times H_2O$ and $1 \times 60\%$ isopropanol. After washing, cells were stained with Oil Red O for 20 min. After staining, cells were washed with $5 \times H_2O$, and ready for viewing under microscope. For Oil Red O quantification of lipid accumulation, Oil Red O-stained cells were washed with $3 \times 60\%$ isopropanol, then stained Oil Red O were extracted by applying 100% isopropanol to the cells for 10 min. Intensity of extracted Oil Red O was measured by the absorbance at 492 nm.

**Quantitative real-time RT-PCR**. Total RNA was purified using RNeasy Plus kit (Qiagen). 10–500 ng RNA was subjected to first-strand cDNA synthesis using Maxima™ H Minus cDNA Synthesis Mastermix (Thermo Scientific). Gene expression was analyzed by quantitative real-time PCR in 20 µl reactions with SYBR Green (Cat#4367659, Applied Biosystems by Thermo Fisher Scientific). All gene expression levels were first normalized with 18 s RNA as an internal control and then were normalized with the level of vehicle (DMSO) treated cells. Primers used in real-time PCR are as following: 18 S, forward, agtccctgcccttttgtacaca, reverse, cgatccg agggcctcacta; human Glut4, forward, TCTTCGAGACAGCAGG GGTA, reverse, CTCCACCAACAACACCGAGA; mouse Glut4,

forward, GCCCCACAGAAGGTGATTGA, reverse, GAAGAT GGCCACGGAGAGAG.

**MRC-5 cell culture and assays.** Human lung fibroblast MRC-5 cell (CCL-171) was purchased from American Type Culture Collection (ATCC) and was cultured in EMEM containing 10% fetal bovine serum, 100 U/mL of penicillin and 100 µg/mL of streptomycin according to manufacturer's instruction. On day 1, cells were seeded in 24-well plates at density of 50,000 cell/well in 1 ml culture medium; On day 2, cells was switched to starve EMEM containing 0.5% FBS; On day3, cells were treated with indicated concentration of ROCK inhibitors, incubated for 1 h, and then TGF-β1 (2.5 ng/mL) was applied to the cells; On day 5 (48 h after treatment), cultured medium was collected for Col1α1 ELISA (human Pro-Collagen 1α1 DuoSet ELISA kit, R&D systems), cells were collected for RNA extraction or for western analysis. Primer's sequence used for real-time PCR are as following: αSMA, forward, CCCAGAC ATCAGGGGGGTGAT, reverse, TCGGGTACTTCAGGGTCAGG; CTGF, forward, CGCACAAGGGCCTATTCTGT, reverse, GAGC ACCATCTTTGGCGGT; Col3A1, forward, CAGCCTCCAACT GCTCCTAC, reverse, CCAGGGTCACCATTTCTCCC.

**siRNA Knock down in Human subcutaneous pre-adipocytes.** ON-TARGETplus SMARTpool siRNA that specifically targeting ROCK1 (L-003536-00-05), ROCK2 (L-004610-00-05) and control siRNA (D-001810-10-05) were purchased from Dharmacon (Thermo Fisher Scientific Inc.). Transfection was performed using Amaxa™ P1 Primary Cell 4D-Nucleofector™ X kit L with Amaxa™ 4D-Nucleofector™ X unit (Amaxa Biosystems, Lonza) according to manufacturer's instructions. Briefly, cells were trypsinized from culture plates, $5 \times 10^5$ cell were used for each transfection in 100 µl Single Nucleocuvette. After transfection, cells were plated and recovered for 6 h in incubator before changing to culture medium. Cells were collected for Western blotting analysis 48 h after transfection.

**Preparation of whole cell/tissue extracts and Western blots.** At the time of collection, PBMCs and monocytes cultures were harvested, while MDM were gently scraped in cold PBS after medium removal. Cells or homogenized tissues were centrifuged at 400 g for 5 min at 4 °C, and cell pellets were lysed in 50 mM Tris pH 7.9, 300 mM NaCl and 1% Triton X-100 supplemented with protease and phosphatase inhibitors cocktail (Thermo-Scientific #A32961) for 15 min on ice. After centrifugation at 17,000 g for 10 min at 4 °C, whole cell extracts were collected, protein concentrations were determined using Bradford method (ThermoScientific #23238) and samples were reduced and denatured in DTT-containing XT Sample buffer (Biorad #1610791) for 10 min at 95 °C.

Proteins were resolved by SDS-PAGE (Invitrogen mini gels system) and transferred onto nitrocellulose membranes (Invitrogen iBlot 2) before blocking with 10% evaporated milk in PBS 0.2% Tween and incubation with the corresponding primary and HRP-conjugated secondary antibodies, all diluted in PBS 0.2% Tween 10% evaporated milk at concentrations recommended by manufacturer. Signals were visualized using the Amersham ECL, Pierce ECL Plus or SuperSignal West Pico Plus ECL detection reagents and imaged using Invitrogen iBright CL1500 imager. Red Ponceau staining was used before immunoblotting to check for transfer efficiency. All blots were probed for β-actin as a loading control. When indicated, Western Blot quantification was performed using Fiji/ImageJ software.

**Antibodies.** For Western blots, primary and HRP-conjugated secondary antibodies were obtained 1) from Cell Signaling Technology: anti-β-Actin (4970), anti-Akt (2920), anti-Phospho-Akt S473 (4060), anti-AMPK (2793), anti-Phospho-AMPK T172 (50081), anti-Cofilin S3 (5175), anti-Phospho-Cofilin (3313), anti-p70 S6K (34475), anti-Phospho-S6K T389 (9205), anti-mTOR (2972), anti-Phospho-mTOR S2448 (2971), anti-α-Smooth Muscle Actin (SMA; 56856), anti-STAT3 (12640), anti-Phospho-STAT3 Y705 (9145), anti-Phospho-STAT5 Y694 (4322), HRP-conjugated anti-Rabbit IgG (7074), and HRP-conjugated anti-Mouse IgG (7076); 2) from Sigma Aldrich: anti-ROCK1 (HPA007567) and anti-ROCK2 (HPA007459); or 3) from eBioscience: CD185 (CXCR5) (12-1859-42), CD279 (PD-1) (11-9969-42), Foxp3 (12-4777-42), CD4 (11-0049-42) and Fixable Viability Dye (65-0866-14).

**ELISA.** Measurement of cytokines concentrations was performed using ELISA kits or antibodies 1) from R&D Systems: hCXCL13 (DY801), hIL-1b (DY201), hIL-10 (DY217B), hIL-17 (DY317), hIL-23 (DY1290), hIL-6 (DY206), hPro-Collagen1α1 (DY6220), hTNF (DY210), mIL-6(DY406), mTNF(DY410) or 2) from eBioscience: hIL-21 capture (14-7219-82) and detection (13-7218-81) antibodies, according to the manufacturer's recommendations. Briefly, 96 W Immulon plates were coated overnight with respective capture antibodies diluted at the recommended concentrations in PBS, washed 4 times and blocked with PBS 5% BSA for 1 h before adding the cell supernatants either pure or diluted in appropriate cell medium. Serial dilutions of recombinant cytokines were made on each plate to serve as concentration standard. Incubation with supernatants was done for 2–3 h at RT or overnight at 4 °C with gentle shaking. Plates were then washed 4 times with PBS 0.2% Tween before incubation with respective detection antibodies for 2–3 h at RT with gentle shaking. After washing 4 times, reaction was initiated by adding Ultra-TMB (ThermoScientific #34029), stopped adding a solution of 0.18 M $H_2SO_4$, and absorbance at 450 nm was measured using 540 nm as reference absorbance. Exact cytokine concentration was calculated by using standard curve. When indicated, cytokine concentrations were normalized to the condition with stimulus alone (set at 100) to compare independent samples purified from independent blood donors.

**Statistical analysis and reproducibility.** Different statistical tests were performed to analyze experiments, as indicated: Unpaired t-test or one way ANOVA followed by multiple comparisons Sidak tests to allow two-by-two comparisons. Significance is indicated on each figure as follows: ns, not significant, $*p \leq 0.05$; $**p \leq 0.01$; $***p \leq 0.001$; $****p \leq 0.0001$. All in vitro data represent at least 3 repeats unless otherwise stated.

**Reporting summary.** Further information on research design is available in the Nature Portfolio Reporting Summary linked to this article.

## Data availability

Numerical source data underlying all graphs can be found in Supplementary data files 1 and 2. Uncropped blots can be found in Supplementary data 3. Additional datasets analysis generated during this study are available on reasonable request.

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

## Acknowledgements

The authors thank Dr. Rui Wu for providing GV101. B.R.B. is supported in part by NIH R37 AI34492, P01 H158505 and P01 AI 056299.

## Author contributions

A.Z.-Z., W.C., J.M., M.S.N. M.G., and S.D.W. conceived and designed the study and experiments; A.Z.-Z., W.C., J.M., M.S.N., I.Z., R.M., and C.S. performed experiments; A.Z.-Z., W.C., J.M., M.S.N., I.Z., R.M., M.G., and C.S. analyzed experiments; A.Z.-Z., W.C., J.M., M.S.N., R.M., and C.S. performed statistical analysis; K.M., B.R.B., M.P., and S.D.W. provided constructive advice, expertise or reagents; A.Z.-Z., W.C., and J.M. wrote the manuscript. A.Z.-Z., W.C., J.M., K.M., B.R.B, M.P., and S.D.W. edited the manuscript.

## Competing interests

The authors declare no competing interests. A.Z.-Z., W.C., J.M., and M.S.N. were previously employees of Equilibre Biopharmaceuticals Corporation and performed part of the work for Graviton Biosciences under the shared service agreement between these two companies.
