## [Peer Review File · Communications Biology]

Reviewers' comments:

Reviewer #1 (Remarks to the Author):

This study investigated the impact of ROCK2 inhibitor on liver fibrosis in in vitro cultured cells and in vivo animal models. This manuscript can be further improved by addressing the concerns below.

The authors could show the level of hepatic ROCK2 expression in animals treating ROCK2 inhibitor GV101. In addition, they need to present ROCK2 levels in cultured cells treating ROCK2 inhibitor.

It would be informative if the authors could show the metabolic phenotypes of the Western Diet group with TAA treatment, including body weight and adiposity (fat depot weights). The question is whether the metabolic phenotype of GV101 treatment on the liver is due to ROCK2 inhibition or decreased body weight. Based on leptin data, the body weight of these mice will be reduced by GV101 therapy. In addition, because GV101 suppresses adipogenesis in adipocytes in vitro, adiposity in GV101-treated mice should be decreased.

The quality of Oil Red O staining is poor. The authors need to provide a high-quality image of Red Oil staining.

The results of Fig. 4b and 4d are not strong. The authors may need to quantify Western blots and show bar graph for each molecule. In particular, the reviewer can't find any differences in Fig. 4d .

The focus of this study should be on the impact of GV101 but not compare it with KD 025. Thus, the authors need to rewrite this point throughout the manuscript. The specificity of GV101 can be addressed.

The authors should consider rewriting the final paragraph of the introduction, as it primarily details the observations and does not serve as an abstract.

The results section can be shortened significantly. The introduction of some result sections is very long. This is not a review paper. This study is very simple and straightforward.

Reviewer #2 (Remarks to the Author):

General Comments:

Zanin-Zhorov and co-authors present a study demonstrating the therapeutic efficacy of selective ROCK2 inhibition in vivo two liver fibrosis models and in vitro systems. They use a compound, GV101, which has 1000-fold selectivity over ROCK1, and administer it to mice after TTA + high fat diet induced liver fibrosis. They demonstrate in vitro decreased ROCK2 induced signaling pathways (pSTAT3, pcofilin) and increased pSTAT5 and pAMPK. They show these findings are mediated through reduced fibroblast activation, adipogenesis, and TLR mediated cytokine secretion by innate immune cells. Prior work by this group and others have shown although there is 90% homology between ROCK1 and ROCK2, only ROCK2 regulates proinflammatory cytokines IL-17 and IL-21 through STAT3 leading to regulation of the balance between pro-inflammatory and immunosuppressive subsets of T cells. GV101 was compared to KD025 in a therapeutic experiment (weeks 6-12), performing better and demonstrating potential reversal of fibrosis. KD025 had toxicities in these studies at high doses and had to be reduced. GV101 appeared to be more effective than KD025 in blocking fibroblast activation and blocking IL-17 from T cells. They also saw effects on Kupffer cells (monocyte/macrophage cell population). These studies show a strong anti-inflammatory effect with GV101 on innate immune cells in the liver to prevent fibrosis. Target that hits metabolism,

inflammation, and fibrosis. There are several strengths of this study, including the therapeutic strategy of compound, comparison to other inhibitors, and mechanistic studies. A few comments remain, but overall, the studies are well done, and the manuscript is well written.

Major Comments:

1. It is interesting that GV101 worked so well in one model (TAA) in a dose dependent manner but not the other model (Western diet). Some discussion on this would be helpful- differences in the model are noted such as in WD, pSTAT3 is not upregulated, and with the high dose GV101, pcofilin increases. Why do the authors think this happens?
2. The authors do not look at pMLC as a marker of ROCK2 activity, which raises the question of whether that is mainly a marker of ROCK1 activity (stress fibers) or not affected in these models. Would be helpful to comment on the absence of this frequently used ROCK readout.
3. The authors seem to base the cells of interest on prior literature, but there is no mention of the effect of ROCK2 inhibition on the vascular compartment in these models. Would be helpful to comment on whether ROCK2 inhibition impacts endothelial cells in these models, in addition to immune cells and fibroblasts. While not the focus of this manuscript, it would be useful to explain why it was not explored.

Minor Comments:

1. Graphs of OHP should show # of mice as well as bars and stats.
2. All Western blots should include quantification with ratio of density of band of interest / control or standard housekeeping protein.

Reviewer #3 (Remarks to the Author):

This paper tests the impact of a selective new ROCK 2 inhibitor, GV101, on liver fibrosis using two different mouse models of liver fibrosis : TAA toxic chronic injury with normal or western diet. The experiments are well performed. However, concerning the TAA model, it does not really present new data, another ROCK2 inhibitor, KD025, having already been showed to ameliorate liver fibrosis using the same experimental approach (ROCK2 inhibition attenuates profibrogenic immune cell function to reverse thioacetamide-induced liver fibrosis. Nalkurthi C et al JHEP Rep. 2021 Oct 6;4(1):100386. doi: 10.1016/j.jhepr.2021.100386. eCollection 2022 Jan. PMID: 34917911). Concerning the Western diet + TAA model, both Rock inhibitors GV101 and KD025 gave very different results in terms of mortality and inflammation. Lethality observed with the used dose of KD025 is not understood from the given experiments and forces authors to reduce the dose during the experiment hampering a real comparison. To understand the mechanisms involved the authors used on the one hand PBMC stimulated by LPS and on the other hand liver Kupffer cells from human livers in the presence of various concentrations of GV101 and showed a reduction in various pro-inflammatory and/or profibrogenic cytokines. The same analysis should have been carried out in mice for Kupffer cells. A deeper analysis should have been given to conclude that the inflammation was responsible for this discrepancy. It should have been for example interesting to compare both inhibitors since with KD025 the impact on fibrosis is modest and there is still a reduction in IL17, suggesting that IL17 is not involved in the anti-fibrotic answer. Finally, the study on adipocytes seem rather disconnected from the initial question on the role of ROCK2 inhibition on fibrosis since it is not shown that it has participated to the observed phenotype.

Other remarks :

- Significance is not indicated for KD025, particularly between GV101 and KD025 in figure 1
- It seems that there is an opposite correlation between the GV101 dose and fibrosis the lowest dose having the highest impact. How do the authors explain it ?

- The quantification of picrosirius red should be given in figure 1 and 2.
- What is the level of inflammatory markers in the total liver of these animals in figure 1 (TNF, IL1, IL6, IL17, CCL2...) . Same question for TNF and IL6 in figure 2
- What is the cause of the lethality in KD025 treated animals ?
- How the authors explain the absence of impact on fibrosis in this high fat diet context for KD025 ?
- Fig3d,e asma expression a quantitative analysis with statistical comparison should be given
- There is no Supplementary figure e.
- How many different samples were used for IL17 , IL21 and CXCL13 expression in human T cells ? does Suppl fig 4 a show 2 representative different samples ?
- What about Kupffer cells in the TAA + Western diet mouse liver context for IL6 and TNF production with and without GV101/KD025?

RESPONSE TO REVIEWERS

Manuscript

Selectivity matters: Novel ROCK2 inhibitor ameliorates established liver fibrosis via targeting inflammation, fibrosis, and metabolism

We are happy that the Reviewers found our manuscript “**well written**”, presenting “**several strengths**” including experiments that are “**well performed**” and demonstrate “**a strong anti-inflammatory effect of GV101**” in the liver “**to prevent fibrosis**”.

We also thank the 3 Reviewers for their critical comments and suggestions, which allowed us to improve the manuscript, ameliorate our data and overall strengthen our conclusions.

We provide below a point-by-point response (marked in blue) to the insightful queries raised by the Reviewers.

REVIEWER 1

This study investigated the impact of ROCK2 inhibitor on liver fibrosis in *in vitro* cultured cells and in vivo animal models. This manuscript can be further improved by addressing the concerns below.

- 1) The authors could show the level of hepatic ROCK2 expression in animals treating ROCK2 inhibitor GV101. We followed Reviewer 1’s recommendation and we have now included the ROCK2 expression by Western blots, both in Livers and Spleens in *Supplementary Figures 1f* and *2f*.

These Western blots do not show any significant variation of the ROCK2 levels in the different conditions, as expected.

- 2) They need to present ROCK2 levels in cultured cells treating ROCK2 inhibitor.

We also included the Western blots for ROCK2 protein in various experiments: T cells (*Supplementary Figure 4c*), PBMCs (*Supplementary Figure 5b*), Monocytes (*Supplementary Figure 5e*), MDM (*Supplementary Figure 5i*) and human Kupffer cells (*Figure 4e*).

As expected, in all these cell types, ROCK2 levels were not changed by the treatment with ROCK2 inhibitors KD025 or GV101.

- 3) It would be informative if the authors could show the metabolic phenotypes of the Western Diet group with TAA treatment, including body weight and adiposity (fat depot weights).

As requested by Reviewer 1, we have now added a graph showing the body weight of the different groups of mice in *Supplementary Figure 2C* of the revised manuscript. The adiposity itself was not measured during the study.

These data do not show any significant variation of the mice body mass, except for the group treated with KD025 150mg/kg as this dose showed toxicity, as we discussed in the manuscript.

- 4) The question is whether the metabolic phenotype of GV101 treatment on the liver is due to ROCK2 inhibition or decreased body weight. Based on leptin data, the body weight of these mice will be reduced by GV101 therapy.

As Reviewer 1 suggests, a decrease in leptin levels can be correlated with a loss in adiposity, which eventually can result in loss of body weight. However, this correlation is not necessarily observed at any given time. Here for instance, the body weights (which is linked to adiposity) did not change with GV101 treatment, yet we observed reduced levels of plasma leptin levels. Therefore, we believe that the beneficial metabolic changes observed in treated mice are due to ROCK2 inhibition *per se*, and not to changes in body weight. Efficiency of ROCK2 inhibition was demonstrated by robust inhibition of cofilin (ROCK2 target) phosphorylation in tissue (*Figure 2e*). However, we do not exclude the possibility that with prolonged treatment of GV101, the body weight/adiposity will be decreased and that will further benefit the whole-body metabolic phenotype.

- 5) In addition, because GV101 suppresses adipogenesis in adipocytes in vitro, adiposity in GV101-treated mice should be decreased.

Reviewer 1 is right: we show that GV101 suppresses adipogenesis in vitro. However, we haven't assessed the adiposity of mice in our present study, but we believe it will be a good point to test in our future studies. The size of adipose tissue depends not only on the number of adipocytes but also on the size of these adipocytes. In the present study we did not observe body weight change after GV101 treatment of 12 weeks, we can hypothesize the adiposity in GV101-treated mice might not be decreased. Again, we do not exclude the possibility that with extended treatment of GV101, the body weight/adiposity will be decreased, and it will be interesting to find out in future studies.

- 6) The quality of Oil Red O staining is poor. The authors need to provide a high-quality image of Red Oil staining.

We followed the suggestion of Reviewer 1, and we have now added high quality and higher magnification images in the revised manuscript (*Figure 3a*). These images allow us to observe more precisely the Oil Red O staining in adipocytes. We have yet kept the non-magnified pictures of the cells in *Supplementary Figure 3a*, as these images have been used for the signal quantification depicted in *Figure 3a*.

7) The results of Fig. 4b and 4d are not strong. The authors may need to quantify Western blots and show bar graph for each molecule. In particular, the reviewer can't find any differences in Fig. 4d.

As requested by Reviewer 1, we have now provided quantification for these specific Western blots as shown in *Figure 4b* and *Supplementary Figure 5h* (formerly *Figure 4d*), as well as for all Western blots included in the manuscript.

These quantifications show a neat decrease in inflammatory and anabolic pathways with GV101 in PBMCs (*Figure 4b*), and a slight decrease in these same pathways as well a slight increase in catabolic AMPK in MDM (*Supplementary Figure 5h*, formerly *Figure 4d*), confirming our conclusions.

8) The focus of this study should be on the impact of GV101 but not compare it with KD025. Thus, the authors need to rewrite this point throughout the manuscript. The specificity of GV101 can be addressed.

The specificity of GV101 is mentioned in the main text and highlighted with the detailed list of kinases not targeted by the compound provided in the Materials and Methods section (page 29). In addition, we emphasized the impact of GV101 on inflammatory, fibrotic and metabolic signaling pathways throughout the manuscript as it was suggested by Reviewer 1. However, we believe that the comparison with KD025 (a previously characterized inhibitor of ROCK2 with less selectivity for ROCK2 compared to GV101) remains **crucial to emphasize the importance of ROCK2 selectivity** in the anti-inflammatory, anti-fibrotic and metabolic effects of GV101.

9) The authors should consider rewriting the final paragraph of the introduction, as it primarily details the observations and does not serve as an abstract.

We have followed Reviewer 1 advice and have substantially modified the final paragraph of the introduction (page 4 and 5). It is now more concise and focuses on the key conclusions of our study.

10) The results section can be shortened significantly. The introduction of some result sections is very long. This is not a review paper. This study is very simple and straightforward.

The results section of the revised manuscript was significantly shortened, with a particular effort made in the beginning of the results section around the innate immune cells (*Figure 4* and *Supplementary Figure 5*)

REVIEWER 2

General Comments:

Zanin-Zhorov and co-authors present a study demonstrating the therapeutic efficacy of selective ROCK2 inhibition in vivo two liver fibrosis models and in vitro systems. They use a compound, GV101, which has 1000-fold

selectivity over ROCK1, and administer it to mice after TTA + high fat diet induced liver fibrosis. They demonstrate in vitro decreased ROCK2 induced signaling pathways (pSTAT3, pCofilin) and increased pSTAT5 and pAMPK. They show these findings are mediated through reduced fibroblast activation, adipogenesis, and TLR mediated cytokine secretion by innate immune cells. Prior work by this group and others have shown although there is 90% homology between ROCK1 and ROCK2, only ROCK2 regulates proinflammatory cytokines IL-17 and IL-21 through STAT3 leading to regulation of the balance between pro-inflammatory and immunosuppressive subsets of T cells. GV101 was compared to KD025 in a therapeutic experiment (weeks 6-12), performing better and demonstrating potential reversal of fibrosis. KD025 had toxicities in these studies at high doses and had to be reduced. GV101 appeared to be more effective than KD025 in blocking fibroblast activation and blocking IL-17 from T cells. They also saw effects on Kupffer cells (monocyte/macrophage cell population). These studies show a strong anti-inflammatory effect with GV101 on innate immune cells in the liver to prevent fibrosis. Target that hits metabolism, inflammation, and fibrosis. There are several strengths of this study, including the therapeutic strategy of compound, comparison to other inhibitors, and mechanistic studies. A few comments remain, but overall, the studies are well done, and the manuscript is well written.

We thank Reviewer 2 for his/her appreciation of our manuscript. We're happy to respond to his/her specific comments below.

Major Comments:

- 1) It is interesting that GV101 worked so well in one model (TAA) in a dose dependent manner but not the other model (Western diet). Some discussion on this would be helpful- differences in the model are noted such as in WD, pSTAT3 is not upregulated, and with the high dose GV101, pcofilin increases. Why do the authors think this happens?

Reviewer 2 raises a good point. The efficacy of GV101 in both models correlates with peripheral exposure of the drug. The levels achieved with 150 mg/kg dose in TAA model are equivalent to levels reached with 30 mg/kg dose of GV101 in TAA+WD as shown in *Supplementary Figure 2b*, pointing to a potential food effect on peripheral exposure of GV101. The results in *Figure 2* suggest that the efficacy of GV101 treatment in TAA+WD achieved a plateau in some of the measured readouts including levels of Hydroxyproline in the liver (*Figure 2b*), phosphorylation of Cofilin and STAT3 in the liver (*Figure 2e*), levels of IL-17 in the plasma (*Figure 2d*), whereas other parameters were modulated in a dose-dependent manner, such as levels of plasma Leptin, Insulin and IL-1 β (*Figure 2d*) or phosphorylation of AMPK and Akt in the liver (*Figure 2e*). This might reflect the difference in dynamics, nature and location of measured readouts, for example, phospho-proteins in tissue vs cytokine in the periphery.

- 2) The authors do not look at pMLC as a marker of ROCK2 activity, which raises the question of whether that is mainly a marker of ROCK1 activity (stress fibers) or not affected in these models. Would be helpful to comment on the absence of this frequently used ROCK readout.

As Reviewer 2 may know, ROCK kinases directly phosphorylate several targets: MLC, MYPT, cofilin, adducin, extrin/radixin/moesin, calponin, MARCKS, LIM kinase, tau, MAP2, vimentin, CRMP-2 among many others (*Amano et al.*, JCB (2015) Vol 209 No.6: 895-912, PMID: 26101221 ; Crit Rev Biochem Mol Biol Early Online 1-16).

Newell-Litwa et al. (JCB (2015) Vol 210 No.2: 225-242, PMID: 26169356) reported that ROCK2, but not ROCK1 shRNA caused a robust down-regulation of intracellular cofilin phosphorylation, validating pCofilin as a marker for selective ROCK2 inhibition in cytoskeletal remodeling during fibrosis. We have included additional explanation in the text of the revised manuscript.

- 3) The authors seem to base the cells of interest on prior literature, but there is no mention of the effect of ROCK2 inhibition on the vascular compartment in these models. Would be helpful to comment on whether ROCK2 inhibition impacts endothelial cells in these models, in addition to immune cells and fibroblasts. While not the focus of this manuscript, it would be useful to explain why it was not explored. ROCK inhibition has been studied in various models of hepatic fibrosis in animals, however there is no published evidence that ROCK2 targeting impacts endothelial cells in these models. Recent studies point to a critical role that endothelial cells play in regulating essential hepatic metabolic functions (*Kaffe et al.*, Cell 2023, PMID: 37562401)). Furthermore, ROCK2 is known to be the key isoform driving brain endothelial damage (*Niego et al.*, Plos One 2017, PMID: 28510599) and LPA-induced activation in human endothelial cells (*Shimada et al.*, JBC 2010, PMID: 20164172). Therefore, we agree with the reviewer that studying the impact of ROCK2 targeting on vascular compartment, notably in liver fibrosis models, will be critical in future studies. Given the amount of work this would represent, we do not believe this can be done within the current manuscript.

Minor Comments:

- 1) Graphs of OHP should show # of mice as well as bars and stats.
We have followed the suggestion of Reviewer 2 and have now added the number of mice, error bars and statistics in the graphs showing the levels of Hydroxyproline in *Figure 1b* and *Figure 2b*.
- 2) All Western blots should include quantification with ratio of density of band of interest / control or standard housekeeping protein.

As requested by Reviewer 2, we have now added quantifications for all the Western blots of the manuscript. All quantifications validate our previous Western blot-based observations and conclusions.

REVIEWER 3

- 1) This paper tests the impact of a selective new ROCK 2 inhibitor, GV101, on liver fibrosis using two different mouse models of liver fibrosis: TAA toxic chronic injury with normal or western diet. The experiments are well performed. However, concerning the TAA model, it does not really present new data, another ROCK2 inhibitor, KD025, having already been showed to ameliorate liver fibrosis using the same experimental approach (ROCK2 inhibition attenuates profibrogenic immune cell function to reverse thioacetamide-induced liver fibrosis.

Nalkurthi C et al JHEP Rep. 2021 Oct 6;4(1):100386. doi: 10.1016/j.jhepr.2021.100386. eCollection 2022 Jan. PMID: 34917911).

Reviewer 3 is right when mentioning that KD025, a previously identified ROCK2 inhibitor, has been shown to improve liver fibrosis in mice model. However, we believe pursuing research about how selective ROCK2 inhibition impact liver fibrosis is still very relevant both scientifically and clinically: 1) can we understand the mechanisms of action by which ROCK2 inhibition ameliorates liver fibrosis? and 2) can we improve the clinical efficiency of the already existing ROCK2 inhibitors?

Here, we characterize GV101, a novel and highly selective ROCK2 inhibitor (10 times more selective for ROCK2 than KD025), and we show in the present study that GV101 safely and efficiently ameliorates established liver fibrosis in mice in a more potent way than KD025.

Moreover, we demonstrate that the therapeutic effect of GV101 is mediated by previously undocumented triple effect of ROCK2 targeting *in vivo*:

- robust anti-fibrotic effect with reduction of collagen levels,
- marked down-regulation of inflammatory pathways including STAT3 and Akt, and
- beneficial metabolic changes with up-regulation of AMPK in liver and diminished serum levels of leptin, insulin and cholesterol.

We believe that all together, these findings are novel and clinically relevant notably in the context of liver fibrosis.

- 2) Concerning the Western diet + TAA model, both Rock inhibitors GV101 and KD025 gave very different results in terms of mortality and inflammation. Lethality observed with the used dose of KD025 is not understood from the given experiments and forces authors to reduce the dose during the experiment hampering a real comparison. As noted by Reviewer 3, GV101 and KD025 had different effects on mice lethality. The lethality observed in KD025 group can be explained by the very high blood levels of the drug achieved when mice received high fat diet. This phenomenon is consistent with previously reported food effect on peripheral exposure of KD025

(Schueller et al, Clin Pharmacol Drug Dev 2022, PMID: 35238174). The levels of KD025 in TAA+WD model (*Supplementary Figure 2b*) were 5X higher than the therapeutic levels achieved in patients (Belumosudil label at [label \(fda.gov\)](https://www.fda.gov)), potentially causing the weight loss and death of treated mice as well as the mixed results with regards to pro-inflammatory cytokines assessment. On the other hand, GV101 safely and efficiently attenuated established liver fibrosis induced by TAA+WD at much lower peripheral exposure (*Supplementary Figure 2b*) due to high potency and selectivity toward ROCK2 (*Supplementary Figure 1a*).

The observation of these differences confirms the relevance of characterizing novel selective ROCK2 inhibitors such as GV101 and comparing them with the already existing molecules such as KD025.

- 3) To understand the mechanisms involved the authors used on the one hand PBMC stimulated by LPS and on the other hand liver Kupffer cells from human livers in the presence of various concentrations of GV101 and showed a reduction in various pro-inflammatory and/or profibrogenic cytokines. The same analysis should have been carried out in mice for Kupffer cells.

As Reviewer 3 noted, we observed a strong anti-inflammatory effect of GV101 in human PBMCs (*Figure 4a*), primary Monocytes (*Supplementary Figure 5c*) and primary Kupffer cells (*Figure 4c*), uncovering a previously unknown role of ROCK2 selective inhibition in myeloid innate immune cells and liver-resident macrophages. We have now strengthened our observations in human Kupffer cells by adding 3 more independent samples (*Figure 4c*), and by providing Western blot analysis demonstrating potent effect on Akt-S6K anabolic pathway (*Figure 4e*).

Understanding whether this effect is also observed in liver-resident Kupffer cells from mice with experimental liver fibrosis would be indeed important for future studies and would require extensive experimental and technical insights that cannot be performed within our present study. However, we have now added experiments performed in murine primary Kupffer cells showing that GV101 treatment also exerts a strong anti-inflammatory effect in murine Kupffer cells (*Figure 4d*), suggesting that the inflammatory response of mice Kupffer cells *in vivo* might also be decreased upon GV101 treatment.

- 4) A deeper analysis should have been given to conclude that the inflammation was responsible for this discrepancy. It should have been for example interesting to compare both inhibitors since with KD025 the impact on fibrosis is modest and there is still a reduction in IL17, suggesting that IL17 is not involved in the anti-fibrotic answer.

IL-17 contributes to liver fibrogenesis through multiple mechanisms, including direct activation of hepatic stellate cells to promote collagen production. ROCK2 inhibition has direct anti-fibrotic effect in fibroblasts and down-regulates IL-17 secretion in T cells. The effect of KD025 treatment was less robust both *in vivo* (TAA model) and *in vitro* due to limited potency of this compound against ROCK2 in comparison to more potent and selective novel ROCK2 inhibitor, GV101 (*Supplementary Figure 1a*: KD025's ROCK2 IC₅₀ = 287 nM vs GV101's ROCK2 IC₅₀ = 7.5 nM). In TAA+WD model, KD025 treatment was not well tolerated causing the weight loss and death

of treated mice due to very high levels of the drug achieved in the periphery as previously mentioned (*Supplementary Figure 2b*). This might potentially explain the mixed results observed in KD025 group with regards to pro-inflammatory cytokine assessment in serum (*Figure 2d*): IL-1 β was increased, whereas IL-17 levels were reduced compared to the vehicle treated mice.

- 5) Finally, the study on adipocytes seems rather disconnected from the initial question on the role of ROCK2 inhibition on fibrosis since it is not shown that it has participated to the observed phenotype.

As Reviewer 3 may know, adipose tissue and its secreted adipokines are well known to create lipotoxic milieu, promote inflammation and fibrosis in the liver. Therefore, by using primary human pre-adipocytes and murine 3T3L1 cells, we aimed to define the direct effect of ROCK2 targeting on adipogenesis and metabolic pathways that impact hepatic fibrogenesis. In addition, previous studies demonstrated that the anti-adipogenic effect of KD025 is mediated by inhibition of both ROCK2 and CK2 (*Tran et al.*, *Molecules* 2021, PMID: 34443331). The results of present work showed that highly potent and selective inhibition of ROCK2 by GV101 effectively down-regulates adipogenesis which was associated with up-regulation of AMPK phosphorylation, observed also in liver tissues of GV101-treated mice.

Other remarks:

- 1) Significancy is not indicated for KD025, particularly between GV101 and KD025 in figure 1

We have now included the statistical analysis for KD025 versus GV101 in *Figure 1b*.

- 2) It seems that there is an opposite correlation between the GV101 dose and fibrosis, the lowest dose having the highest impact. How do the authors explain it?

The data in Figure 2 suggest that the efficacy of GV101 treatment in TAA+WD achieved plateau in some of the measured readouts including levels of Hydroxyproline, phosphorylation of Cofilin and STAT3, levels of IL-17, whereas other parameters were modulated in a dose-dependent manner, such as levels of plasma Leptin, Insulin and IL-1 β , or phosphorylation of AMPK and AKT. This can reflect the difference in dynamics, nature and location of measured readouts. For example, phospho-proteins in tissue vs cytokine in the periphery.

- 3) The quantification of picrosirius red should be given in figure 1 and 2.

We have now included the quantification in *Supplementary Figures 1b* and *2a*, respectively.

- 4) What is the level of inflammatory markers in the total liver of these animals in figure 1 (TNF, IL1, IL6, IL17, CCL2...). Same question for TNF and IL6 in figure 2.

We conducted PCR analysis to define the expression levels of pro-inflammatory and pro-fibrotic markers in liver tissue, however the generated data was inconclusive due to high sample-to-sample variability and no clear impact

of the disease: normal group was the same as TAA group at W9, as Reviewer 3 can observe in the graph below. Therefore, we did not include this data in the revised manuscript. The inflammatory component in both models was defined by robust increase of STAT3 and Akt phosphorylation in the liver tissue which was diminished by selective ROCK2 inhibition (*Figures 1d, 1e and 2e*).

5) What is the cause of the lethality in KD025 treated animals?

As discussed previously in the response to Reviewer 3's second main comment, the lethality in KD025 group was probably caused by significant weight loss associated with extremely high levels of the drug in the periphery reaching 5X the therapeutic levels in patients.

6) How do the authors explain the absence of impact on fibrosis in this high fat diet context for KD025?

The absence of impact on fibrosis and some of pro-inflammatory parameters in KD025 group can be explained by the fact that animals in this group suffered significant weight loss and struggled to stay alive.

7) Fig3d, e asma expression a quantitative analysis with statistical comparison should be given.

We have now included the quantifications in *Figures 3d and 3e*.

8) There is no Supplementary figure e.

We have double checked to make sure that there is *Supplementary Figure 3e* in the revised manuscript.

9) How many different samples were used for IL17, IL21 and CXCL13 expression in human T cells? does Suppl fig 4 a show 2 representative different samples?

Supplementary Figure 4a shows 2 representative different samples. Experiments conducted with GV101 and KD025 were repeated by using 3 different donors and the data is summarized in the new table in the figure.

10) What about Kupffer cells in the TAA + Western diet mouse liver context for IL6 and TNF production with and without GV101/KD025?

As previously discussed in response to Reviewer 3's 3rd main comment, we have now conducted additional experiments using murine primary Kupffer cells and confirmed that GV101 and KD025 down-regulate IL-6 and TNF production the same way it did in human primary Kupffer cells. The new datasets are included in Figure 2d and Supplementary Figure 5g.

REVIEWERS' COMMENTS:

Reviewer #1 (Remarks to the Author):

The authors addressed all previous comments.

Reviewer #2 (Remarks to the Author):

The authors have addressed all of this reviewer's comments. I would still recommend that authors include all individual data points on graphs, such as OHP, rather than writing n=10 in the bars, as there is value in seeing the spread of the data.

Reviewer #3 (Remarks to the Author):

In this new version, the authors have moderately improved the manuscript in adding required quantifications. Concerning the role of inflammation, they also added an in vitro experiment on mice Kupffer cells showing that KD025 has lower impact than GV101 on inflammatory phenotype of macrophages comforting the higher efficiency. It remains surprising to see such an impact on liver fibrosis without demonstrating in vivo reduction in liver inflammatory markers. They do not bring experimental evidence neither for understanding the lethality induced by KD025 nor to demonstrate the specific role of IL17 in the anti-fibrogenic impact of GV101.